*agronomy*

*Article*

# Phosphorus Behaviour and Its Basic Indices under Organic Matter Transformation in Variable Moisture Conditions: A Case Study of Fen Organic Soils in the Odra River Valley, Poland

**Magdalena Debicka *, Adam Bogacz and Karolina Kowalczyk**

Institute of Soil Science and Environmental Protection, Wrocław University of Environmental and Life Sciences, Grunwaldzka 53 St., 50-357 Wrocław, Poland; adam.bogacz@upwr.edu.pl (A.B.); 105668@student.upwr.edu.pl (K.K.)

* Correspondence: magdalena.debicka@upwr.edu.pl

**Citation:** Debicka, M.; Bogacz, A.; Kowalczyk, K. Phosphorus Behaviour and Its Basic Indices under Organic Matter Transformation in Variable Moisture Conditions: A Case Study of Fen Organic Soils in the Odra River Valley, Poland. *Agronomy* **2021**, *11*, 1997. https://doi.org/10.3390/agronomy11101997

Academic Editors: Bořivoj Šarapatka, Miroslav Dumbrovský and Jana Podhrázská

Received: 1 September 2021
Accepted: 29 September 2021
Published: 1 October 2021



**Abstract:** Lowering of groundwater levels caused by anthropogenic changes in the environment gives rise to global problems, most of which relate to soil degradation such as land desertification or organic soil degradation. The transformation of drainage-sensitive organic soils causes many irreversible changes during organic matter (OM) transformation. Phosphorous (P) behaviour is one of the aspects of OM transformation that requires further investigation, due to the P transformations' complex dependency on many environmental factors. Our study aimed to characterise behaviour of P and find indices reflecting P changes under the influence of OM transformation in drained organic soils in the Odra river valley. The studies were carried out on soils representing different stages of soil degradation in which basic soil properties, including different P forms, were determined with commonly used methods. The results showed significantly higher content of soluble P forms (Pw, $P_{CaCl2}$, $P_{M3}$), particularly in the most drained postmurshic soil (P1). The indices used in this study—Ip, PSD, C:Pt, N:Pt—reflected well the P and OM transformations in organic soils degraded by drainage. This was indicated by numerous statistically significant relationships between the indices and basic soil properties (e.g., Ash, C, N), as well as different P forms (Pt, Pmin, Pox, Porg, Pw, $P_{CaCl2}$, $P_{M3}$). The PSD and Ip values increased and the C:Pt and N:Pt ratios decreased with the degree of OM mineralisation and the degree of site drainage (P3 < P2 < P1).

**Keywords:** drained peatlands; organic soils degradation; organic matter mineralisation; P forms; stochiometric ratios; phosphorus saturation degree; P solubility; P transformations

## 1. Introduction

Organic soils, in particular peat soils, are of unique importance in the natural environment. These are classified as Histosols, namely, soils with at least 40 cm of organic material within the top 80 cm of the soil profile that are composed of at least 20–35% organic matter [1]. These soils cover almost 3% of the global land area [2–4]. Their extent, however, is small considering these soils' enormous importance to the environment on a global scale. These soils bind huge amounts of carbon dioxide ($CO_2$) and prevent it from escaping into the atmosphere, thus representing one of the largest natural carbon reserves. Peatlands are also a main store of water in local and global ecosystems, and, therefore, tend to have a regulating effect on the climate. For these reasons, organic soils play a key role in counteracting climate change, which is one of the most significant problems currently facing human civilisation [5].

Unfortunately, organic soils have been subject to intensive transformation leading to their degradation for many years. This transformation is mainly related to natural or anthropogenic soil drainage [6]. Many peatlands worldwide have been artificially drained through various drainage systems and cleared of vegetation in order to obtain fertile land for agricultural purposes [6–12]. As cited by Grenon et al. [8], of the global peatlands

converted to agriculture Tubiello et al. [4] estimate that 60% are in cool temperate/boreal climates, 34% are in tropical regions and 5% are in warm temperate areas.

Permanent lowering of water level in peat soils leads to a change from anaerobic conditions, which favour the accumulation of organic matter and the formation of peat layers, to aerobic conditions, causing an increase in microbiological activity. This in turn supports the transformation of organic matter by mineralisation and humification leading to quantitative carbon loss and many other important changes for soil functioning, some of which are still not fully characterised or well-understood [8,10,11]. These processes are accompanied by a number of irreversible changes in physical properties, which manifest themselves as changes in morphology, soil structure, deterioration of water properties, as well as changes in chemical and physicochemical properties, associated with the release of biogens during the transformation of organic matter. These changes, known as the moorsh-forming process, lead to a gradual transformation of peat into moorsh and are referred to as secondary transformation of organic matter [5,10–17]. These complex transformations observed after drainage of organic soils lead to irreversible deterioration of peat properties meaning it can no longer fulfil its previous role in the environment; the soils cease to be a carbon store and lose their ability to absorb and store water, hence the above processes are associated with peat degradation [11,18].

Important changes in phosphorus (P) behaviour also take place in drained organic soils [8,17,19–21]. Dehydrated peat soils, which are a "boundary zone" between terrestrial and aquatic ecosystems, are often characterised by excessive P accumulation [17]. Understanding the behaviour of P in the environment is, however, problematic; this behaviour depends on complex and dynamic transformation processes in the soil environment, including mineralisation and immobilisation, dissolution and precipitation, oxidation and reduction, sorption, chelation and other processes [21–26]. Intensive mineralisation of OM and decomposition of organic molecules containing P as a component (Porg) lead to the accumulation of bioavailable and soluble forms of P [21,25,27]. The changes in the oxidation state of Al and Fe ions between anaerobic and aerobic conditions lead to the release of bound P; this process thus becomes another important source of P in soils [17,28], even though Ca plays the main role in P fixation in organic soils [8]. On the one hand, an increase in available and soluble forms of P is responsible for increased soil fertility; on the other hand, the release of P into the environment can have numerous adverse consequences. In arable soils, this threat increases due to excessive P fertilisation and its resultant loss into the environment [8,19,20]. In organic soils, there is also the potential for such a threat, however, this is related more to a change in the P fixation mechanisms. This may improve the nutrient richness of soils, but the threat of eutrophication is also associated with leaching of excessive P in areas adjacent to drained organic soils [29–31].

In an era of accelerating climate change and declining groundwater levels worldwide, various aspects of the transformation of drainage-sensitive organic soils are becoming important issues that require the attention of researchers. The greatest degradation and loss of global mire resources has affected Europe, where losses are estimated at around 52% of their original area [12,32]. However, due to the specific global distribution of organic soils, studies conducted on peatlands in boreal (northern) areas tend to dominate the literature [33]. Far fewer papers have presented results from studies of post-drainage organic matter transformations in fen peatland environments in temperate climates [11,34], which occupy large areas of central Europe and occur mainly in areas of former lakes, ponds and in river valleys [35]. In Poland, peat soils cover 4.07% of the country's total area [36], dominated by lowland peat soils, which account for 92% [37]. The problem of degradation of organic soils is, therefore, the subject of numerous studies, many of which concern organic soils in river valleys on fen peatlands [10,16,17,37,38]. Some studies have also considered P transformations within this soil environment [17,21,22,24–26,39]; these confirm the threat of increased eutrophication of aquatic environments caused by P transformations in organic soils and also highlight the need for improved understanding of P forms' behaviour in organic soils characterised by different habitat conditions and features.

Our study aimed to analyse the behaviour of P forms, particularly soluble ones, and to derive some basic indices reflecting changes occurring due to the influence of organic matter transformation after drainage of organic soils in the Odra river valley. The studies were carried out on organic soils representing different stages of development: from the stage of organic matter accumulation observed in peat soils, through the stage of secondary transformation of organic matter leading to the formation of murshic soils, to the stage of post-murshic mineral-organic soils formed as a result of intensive mineralisation and transformation of organic matter.

## 2. Materials and Methods

### 2.1. Description of the Study Area

The research was carried out in the floodplain of the Odra proglacial valley (sites P2 and P3) and in the area adjacent to the proglacial valley (site P1) near the locality of Przedmoście (51°11′24″ N 16°40′03″ E), situated ca. 25 km north-west of Wrocław, in Lower Silesia (SW part of Poland). The study area and location of study sites are presented in Map 1 (Figure 1).

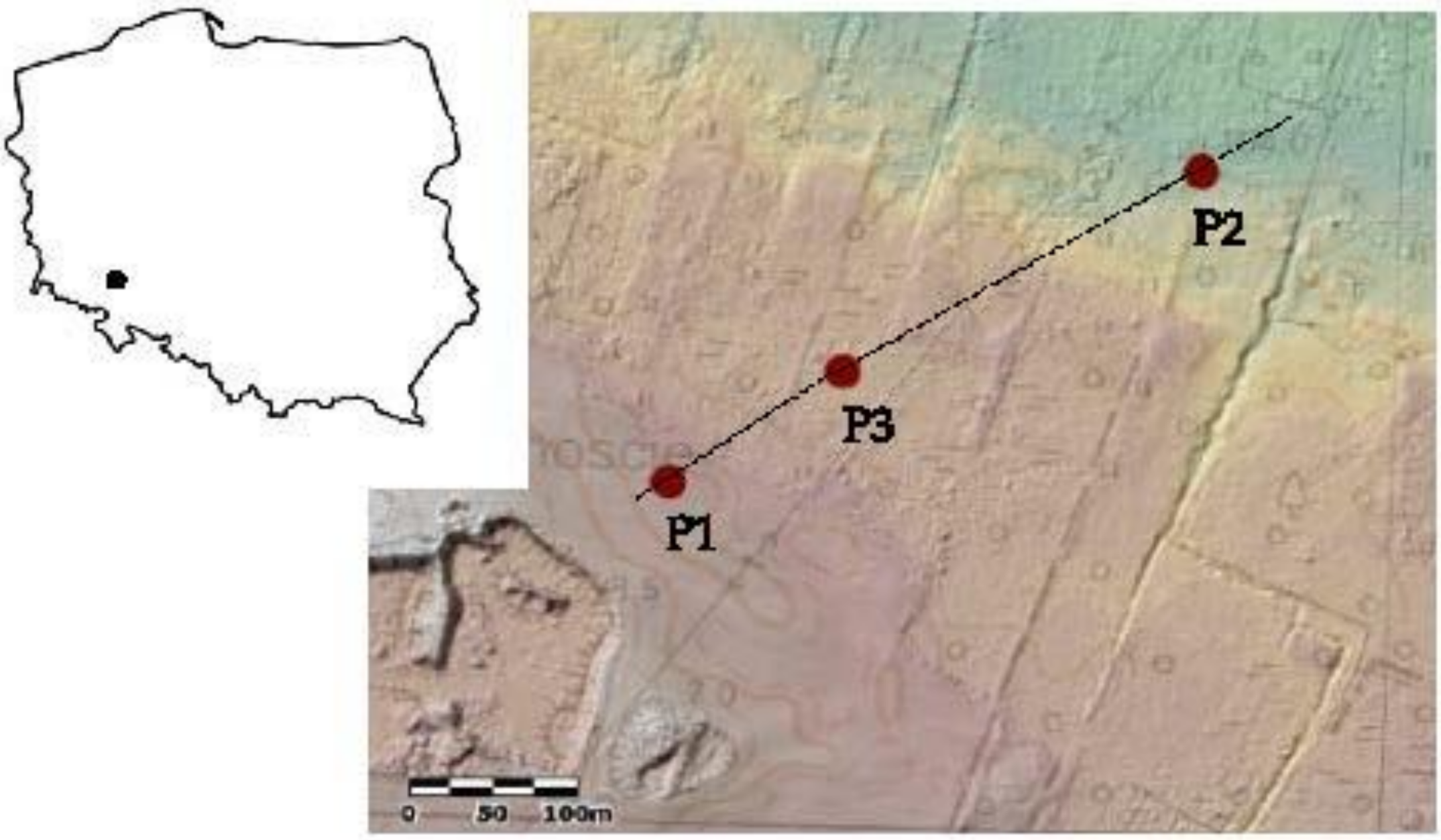


**Figure 1.** Location of the study area and distribution of soil profiles (P1–P3) in the transect.

The hydrographic system of the research area is typical of the coastal part of the river valleys. At its edge, we find areas dried up by natural drainage supported by anthropogenic activities. The soils are supplied with rainwater or short-distance surface transport of water (P1 area). As the terrain descends, we observe the appearance of numerous springs. The evidence of their presence are the spring domes raised up to a few meters above their base. The soils are fed here by the waters of long-distance transport that come out to the surface, often under considerable pressure. A part of the supplying water in this area is related to the system of watercourses currently occurring in the marginal valley surface (P3 area). Moving away from the edge of the marginal valley, the activity of the springs disappears. Then the share of the flowing water increases significantly. Additional water supply is observed in these areas during periodic floods (P2 area). The waters of the source area, which partly supply meadow areas, are alkaline and particularly rich in $Ca^{2+}$, $SO_4^{2-}$ and $PO_4^{3-}$ ions, with relatively low $Cl^-$ content [40].

At this location, the organic formations of the proglacial valley were formed on thin alluvial sands underlain by clays [40]. The organic horizons were originally 90–300 cm thick [41]. Within shallow depressions, lake-related formations initially developed;

these comprise gyttja, then moss peats, reed or alder peats and, finally, sedge peats were formed [42]. These soils (site P2) are usually described as Sapric Eutric Calcic Drainic Murshic Histosols (Limnic) [1]. On the southern side of the study site, there are numerous inaccessible post-peat pits within a distance of several meters. These are currently flooded and overgrown by swamp alder trees in addition to a high proportion of trees of the genus *Salix* spp. [43]. These pits were formed as a result of peat exploitation, which at that time was a valuable fuel resource. The P3 profile was located on one of the post-peat pits, which was classified as Sapric Calcic Eutric Histosols (Limnic) [1].

The area was drained and initial land preparations were made as early as the 1920s. After drainage in the mid-1960s, meadows were established at this site, followed by the construction of a number of major drainage ditches [44]. In the established meadow, plant communities of the classes *Molinio-Arrhenatheretea* and *Carex acutiforni* were found at the end of the 1990s [44]. In the group of grasses, a significant share was occupied by *Festuca rubra* and *Poa pratensis*, which often occur in soils with a clearly marked moorsh-forming process [45]. The diverse composition of meadow flora indicates varied moisture conditions within the study area. Expansion of *Urtica dioica* is also observed, indicating intensive mineralisation of organic matter [46]. This grassland area was used alternately as meadow and pasture until the mid-1990s [43]. Since 2016, the whole area of the grassland has been mown once a year. The investigated meadow areas have previously been characterised in terms of soil science [47], botany [44] and mycology [43]. In analysed organic horizons there were soil fungi species of following genera found: *Acremonium*, *Absidia*, *Mortierella*, *Penicillium*, *Trichoderma*, *Fusarium*, *Mucor* and *Rhizobium.* Fungi had better condition in the meadow habitat due to the lower groundwater level and greater degree of organic matter decomposition. The moorsh process favours the development of fungi and their diversity, especially in the soil surface horizons. Strongly dried horizons were characterised by a particularly large number of species *Penicillium* [43]. During preliminary studies, soil and groundwater levels in meadow areas, depending on the point of measurement, ranged from 20 to 60 cm below ground level [42]. Soil areas P2 and P3 are classified to periodically dry (BC) prognostic soil moisture complex [15].

The area adjacent to the Odra proglacial valley is former farmland comprising post-murshic soils classified as Gleic Umbrisols (Arenic) according to the WRB of Soil Reference Groups [1]; this soil type is represented by soil P1. For many years, fodder crops such as cabbage and lupins were grown on these fields, in addition to rye in the 1990s. Thereafter, the field has not been cultivated, a situation that continues to the present day. This wasteland area is mown once a year and is situated at the highest elevation of this study (Figure 1), hence it is characterised by the lowest levels of groundwater and ground moisture. This area (P1) is classified as dry (C) prognostic soil moisture complex [15].

### 2.2. Methods

The studies were carried out on 3 soils representing different stages of organic soil transformation, which are associated with different moisture conditions. Field studies were conducted in spring–summer time (May–June). Soil morphology was described, and the habitat was characterised as well as the soil material for laboratory testing was collected.

Soil profiles (P1, P2, P3) were located in a transect (Figure 1) crossing a range of habitats. At study sites P1 and P2, soil samples were collected after soil excavations from each genetic horizon. Soil samples at site P3 were taken by means of a 6.0 cm diameter Instorf peat auger (Eijkelcamp Soil and Water, Giesbeek, The Netherlands). Peat cores were sliced according to the genetic horizons.

The collected material was divided into two parts: the first one was placed in a refrigerator at 4 °C for determinations in fresh material; the other was dried at room temperature, then ground and sieved through a 2 mm mesh sieve. Biological debris was not removed from the organic samples. Only in the mineral samples, were plant remnants removed. To obtain a representative sample, fresh material was rolled and then divided (chopped) into equal fragments that were reassembled to repeat this step

several times. In fresh organic material, the degree of decomposition of organic matter was determined using the half-syringe method based on analysis of rubbed and unrubbed fibre content [48], and the SPEC (Sodium Phosphate Extract Colour) method was used to calculate the pyrophosphate index (IP) to determine the degree of peat humification [48]. In all soil samples the measurements of pH were made also in fresh material. pH was determined potentiometrically in $H_2O$ and 1 mol/L KCl and 0.01 mol/L $CaCl_2$ solutions at a soil:solution volume ratio of 1:5. In organic samples, the pH was determined from fresh material. The ash content of the investigated soils was determined by burning them at 550 °C for 4 h; total organic carbon (C) and total nitrogen (N) were determined on a Vario Macro Cube macroanalyser (Elementar Analysensysteme GmbH, Langenselbold, Germany). Before total organic carbon measurements, the inorganic carbon was removed from the samples by treating them several times with 10% HCl (as recommended by the manufacturer). Calcium carbonate content was measured by the Scheibler method [49,50], and hydrolytic acidity (HA) was measured using the Kappen method. Exchangeable base (EB) cation content ($Ca^{2+}$, $Mg^{2+}$, $K^+$, $Na^+$) was extracted with 1 mol/L $NH_4OAc$ at pH 7.0 (1:10 *w/v*) and their concentration in extracts was measured using Microwave Plasma–Atomic Emission Spectrometry (MP-AES 4200 Agilent Technologies, Santa Clara, California, USA). Effective cation exchange capacity (CEC) and base saturation (BS) were also calculated based on the sum of HA and EB. The amounts of non-crystalline and poorly crystalline oxides of Al (Alox), Fe (Feox) and P (Pox) were extracted in darkness with a mixture of ammonium oxalate solution and oxalic acid at pH 3.0 [51], and measured by inductively coupled plasma-atomic emission spectrometry (ICP-OES) using iCAP 7400 Thermo Scientific apparatus (Thermo Fisher Scientific Inc., Waltham, Massachusetts, USA). The P sorption capacity (PSC) and degree of P saturation (DPS) were then calculated according to the following equations [52–55]:

$$PSC = 0.5\ ([Feox] + [Alox]), \quad (1)$$

$$DPS = ([Pox]/PSC) \cdot 100. \quad (2)$$

The following soluble forms of P were determined in the soil material: available P ($P_{M3}$) by the Mehlich III method [56], water-soluble P (Pw) and easily soluble P ($P_{CaCl2}$) extracted using 0.01 mol/L $CaCl_2$ solution [57]. The organic P content (Porg) was estimated using the ignition method [58], modified by Walker, Adams [59] and Shah et al. [60], in which the difference was calculated between the P content (Pt) of soil samples (1 g finely ground soil) after extraction in 50 mL of 0.5 mol/L $H_2SO_4$ in a sample ignited for 2 h at 550 °C (approximate total P content) and in an unignited sample (mineral P content; Pmin). All extracted P forms were measured in replicates using an ICP-OES analyser (iCAP 7400 Thermo Scientific apparatus, Thermo Fisher Scientific Inc., Waltham, Massachusetts, USA). The C:N, N:P, and C:P ratios were calculated from the results. The solubility index of P (Ip), represented as the percentage of water-extractable forms of P (Pw) compared to the total P (Pt) content, was calculated according to equation:

$$Ip = (Pw/Pt) \cdot 100. \quad (3)$$

### 2.3. Statistics

The results obtained were statistically verified using Statistica 13 [61]. Mean values for properties in organic horizons were statistically compared within each soil profile (vertically), as well as between the research sites (horizontally) by the Tukey post-hoc test, at a confidence level of $p < 0.05$. Relationships between the selected parameters were expressed as a correlation coefficient at a statistically significant level of $p < 0.05$. To summarise the results, the analysis of principal components (PCA) was also calculated.

## 3. Results and Discussion

### 3.1. Morphology of Soils

The morphology of the investigated soil profiles is presented in Figure 2 and some details of the basic morphological features are summarised in Table 1.

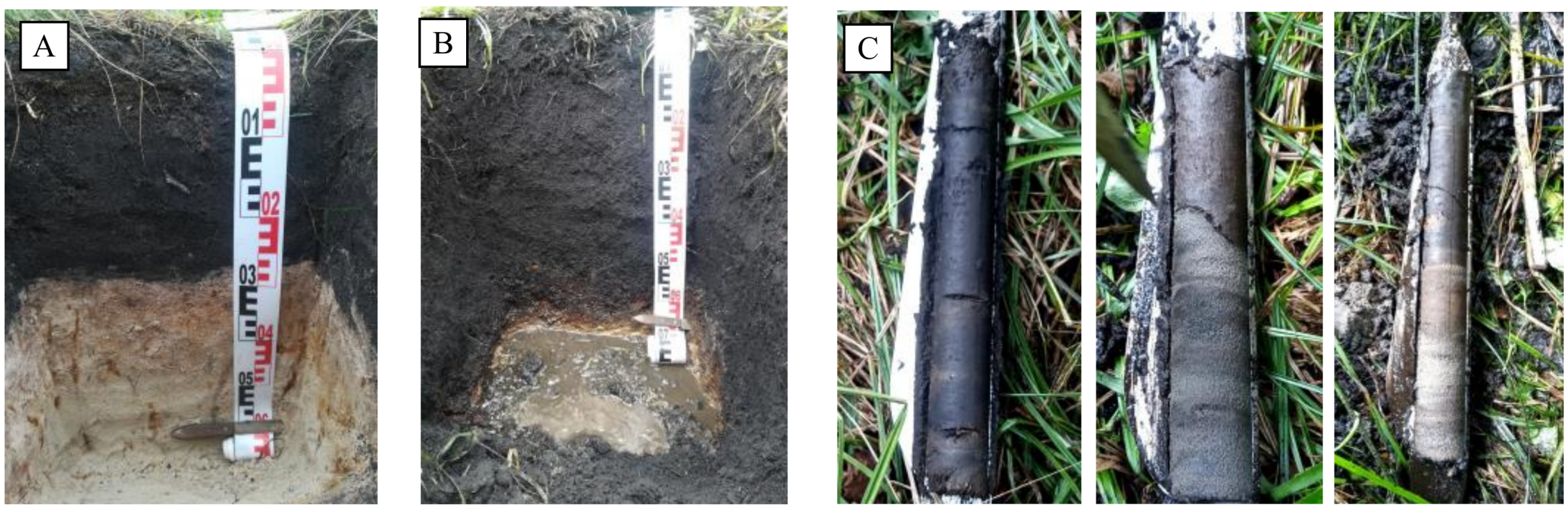


**Figure 2.** Morphology of soils under study: (**A**) Gleyic Umbrisol (Arenic) (postmurshic soil) (P1); (**B**) Calcic Drainic Sapric Murshic Histosols (Limnic) (sapric murshic soil) (P2); (**C**) Calcic Sapric Histosols (Limnic) (sapric peat soil) (P3).

**Table 1.** Basic morphological information of soils.

| Soil Profile | Soil Horizons | Depth (cm) | Munsell Colour Scale | Soil Moisture | Specific Features |
|---|---|---|---|---|---|
| P1 | Gleyic Umbrisol (Arenic) [1]<br>Postmurshic soil [62]<br>GPS coordinates: 51°11′45.5″ N; 16°40′42.7″ E | | | | |
| | Au1 | 0–8 | 10YR 2/2 | Slightly moist | Humic topsoil horizon with labile humus material combined with mineral fraction (quartz); granular structure; reductomorphic and oxymorphic stains; Fe precipitates (fine, 10% coverage); roots moderately fine; smooth boundary |
| | Au2 | 8–18 | 10YR 3/1 | | Humic horizon with labile humus material combined with mineral fraction (quartz); granular structure; roots fine; smooth boundary |
| | Au3 | 18–30 | 10YR 3/1 | | Humic horizon with labile humus material combined with mineral fraction (quartz); granular structure; roots fine; abrupt boundary |
| | Cg1 | 30–45 | 10YR 6/5 | | Mineral horizon; subangular structure, oxymorphic colours 5 YR 4/5; Fe precipitates (medium-fine, 20–30% coverage); smooth boundary |
| | Cg2 | >45 | 10Y/R 7/3 | | Mineral horizon; subangular structure; oxymorphic colours 5 YR 4/5; Fe precipitates (medium-fine, 10% coverage) |

**Table 1.** *Cont.*

| Soil Profile | Soil Horizons | Depth (cm) | Munsell Colour Scale | Soil Moisture | Specific Features |
|---|---|---|---|---|---|
| P2 | Calcic Drainic Sapric Murshic Histosols (Limnic) [1]<br>(thin) murshic soil [62]<br>GPS coordinates: 51°11′52.2″ N 16°40′58.8″ E | | | | |
| | M1 | 0–10 | 10YR 2/1 | Moist | Moorsh horizons; fine and medium granular structure; roots of medium thickness and highly abundant; numerous earthworms; smooth boundaries |
| | M2 | 10–25 | 10YR 2/1 | Moist | |
| | M3 | 25–42 | 10YR 2/1 | Moist | Moorsh horizon; blocky structure; small roots and few in number; earthworms few in number; clear boundary |
| | Oa | 42–55 | 10YR 1/1 | Wet | Sapric sedge peat horizon; fibre amorphous structure; abrupt boundary |
| | Lcca | 55–70 | 7.5 YR 5/6 | Wet | Limnic horizon with Fe and gypsum with wood pieces, Fe precipitates (fine-medium, 60% coverage); calcium carbonate present; single, fine roots; abrupt boundary |
| | Cgg | >70 | 7.5 YR 4/3 | Wet | Mineral horizon with gley reductomorphic colours; Fe precipitates (fine-medium, 60% coverage) |
| P3 | Calcic Sapric Histosols (Limnic) [1]<br>Sapric peat soil [62]<br>GPS coordinates: 51°11′47.9″ N 16°40′47.9″ E | | | | |
| | Oe | 0–14 | 10YR 2/2 | Wet | Hemic sedge peat topsoil horizon; fibrous structure; abrupt boundary |
| | Oa1 | 14–34 | 10YR 2/1 | Wet | Sapric reed and sedge peat; silted; amorphous structure; single plant fragments; smooth boundary |
| | Oa2 | 34–62 | 10YR 2/1.5 | Wet | Sapric reed and sedge peat; amorphous structure; single plant fragments, smooth boundary |
| | Oa3 | 62–70 | 10YR 2/2 | Wet | Sapric sedge peat; amorphous structure; single plant fragments; smooth boundary |
| | Oa4 | 70–83 | 10YR 2/1 | Wet | Sapric sedge peat amorphous structure; single plant fragments; abrupt boundary |
| | Lc | 83–106 | 10YR 3/3 | Wet | Limnic horizon—gyttja with bryales peat fragments and sand; sulphur content; abrupt boundary |
| | Cgg | 106–120 | 10YR 4/1 | Wet | Mineral horizon—sand with gyttja, gleization |

Profile 1 (P1) is a mineral-organic soil, located at the highest position in the study area, in a drained, periodically flooded area. Morphologically, the organic layer of this soil reaches a thickness of 30 cm and the organic material is heavily rotted with a fine weak tubercular structure showing the characteristics of postmurshic soil according to the Polish Soil Classification [62], whereas according to the FAO-World Reference Base it would be classified as a Gleyic Umbrisol (Arenic) [1]. This mineral-organic soil (P1 site) is now transformed into extensive meadow. In the past (10 years ago), *Brassica oleracea*, *Lupinus luteus* and *Secale cereale* were grown here. Due to the high degree of degradation and dryness, the soil is classified as very poor grassland (class VI). Currently, mainly for economic reasons, it is not possible to change the hydrological conditions in this area. In the case of arable usage, these soils obtain the V bonitation class and are considered as poor (according to the Polish bonitation system).

Profiles 2 (P2) and 3 (P3) are organic soils formed from sedge peat. Soil 2 (P2) is situated at the lowest location in the surveyed transect (Figure 1) and occurs in a drained area cut by numerous drainage ditches. This area is periodically inundated by water; under normal conditions, this soil remains flooded in the lower and middle part of the profile, as indicated by numerous reductomorphic features accompanying gleization in this part of the profile. In the upper part of the profile, the water level shows considerable and frequent fluctuations, which created favourable conditions for the aerobic transformation of organic material to form the moorsh horizon (Figure 2B). In the deeper part of the profile ferruginous gyttja occurs along with fragments of wood, accompanied by calcium carbonate (including the skeletons and shells of lake organisms) indicating the limnic origin of this soil and the waters supplying it. Soil 2 is a murshic soil, classified as a Calcic Drainic Sapric Murshic Histosol (Limnic) according to FAO-WRB [1]. The murshic soil is moderately decayed and was used as a meadow or pasture even 20 years ago [43]. Currently, it is mowed once a year. The level of soil and groundwater usually does not fall below 50 cm below the ground level. For proper use of the meadow, it should not fall below 30 cm below the ground level. Such a level of hydration limits the degradation process of these soils. As a greenland, with appropriate regulation of the water level, it would be classified under class III—grassland of medium quality. In the case of arable use, these soils would obtain the IVb bonitation class (according to the Polish bonitation system) and are considered to be of medium quality.

Soil 3 (P3) is located in a field a few meters higher than the murshic soil (P2). Soil 3 is a fen peat soil, formed from sedge-cane peat with muds present in some horizons. It is supplied by groundwater seeping to the surface. In the lower part of the profile, gyttja is mixed with elements of moss peat and sand, indicating the deposition of these materials by supplying waters was simultaneous with the genesis of organic material. This soil is located in a wet, wooded habitat dominated by species of birch and alder. It is an organic soil classified according to FAO-WRB as Calcic Sapric Histosol (Limnic) [1]. The mineral bed and organic layers within the investigated soils have a sand grain size, as confirmed by laboratory tests. This fen peat is moderately peated. At the beginning of the 20th century, the area was used as a meadow for a short time. From the 1930s, the plant succession of a forest and bush community began. For most of the year, the level of soil water is close to the surface. If the area was re-transformed into a meadow, due to the present strong shrubbery, it would be included in the V class of poor quality grassland. As an arable land, it would be assigned to bonitation class IVa (medium quality; according to the Polish bonitation system). Currently, this facility is excluded from any agricultural use.

### *3.2. Basic Soil Characteristics*

#### 3.2.1. Degree of Peat Decomposition

Both methods used to determine the degree of peat decomposition showed highly consistent results (Table S1). Based on the IP index, the tested organic material was determined as sapric in all tested cases. Based on the fibre content, in only one case was the organic material classified as medium decomposed (R2) hemic type, in the Oe horizon in P3, which represents the most recent soil accumulation. In the remaining samples, the highly decomposed nature of the organic material was confirmed to be (R3) sapric.

#### 3.2.2. Ash Content

Ash content in the studied soils increased with the degree of drainage, thus the highest values were found in P1 and the lowest in P3 (Table 2). Ash content (Ash) in the peat soil (P3) horizons varies from 31.7% to 46.1% of dry matter (d.m.). In the mineral horizons (Lc, Cgg), ash content increases to above 83% of d.m. The variation of ash content in the O horizons within this soil may relate to the varying degree of siltation in the deposited organic layers, which is strongly controlled by the hydrological conditions of this habitat. Murshic soil (P2) shows approximately 2 times higher ash content in its organic horizons compared to values recorded in P3. The ash content ranges from 63% to 76% d.m. in

the organic sediments, including the Lcca horizon containing ferruginous gyttja. In the postmurshic soil (P1), the Ash is much higher, reaching more than 88.6% d.m., indicating a high degree of mineralisation of organic matter in this soil. The variation in Ash values between organic horizon groups is statistically significant as shown in Figure 3A. On the basis of the above results, the soils studied may be arranged by increasing Ash, i.e., peat soil (P3) < murshic soil (P2) < postmurshic soil (P1). This order also, therefore, relates to the extent of organic matter transformation taking place in these soils—from the stage of accumulating organic matter (P3), through the stage of mineralising organic matter and mursh formation (P2), to the stage of intensive mineralisation of organic matter, including its transformation and humification (P1).

**Table 2.** Physicochemical and chemical properties of soil and ash content.

| Number of Profile | Soil Horizons | Depth of Horizon (cm) | pH $H_2O$ | | pH KCl | | pH $CaCl_2$ | | $CaCO_3$ | C | N | Ash |
|---|---|---|---|---|---|---|---|---|---|---|---|---|
| | | | | | | | | | | % | | |
| P1 | Au1 | 0–8 | 5.7 | ±0.09 | 4.6 | ±0.08 | 4.9 | ±0.11 | 0.2 | 4.27 | 0.29 | 88.60 |
| | Au2 | 8–18 | 5.7 | ±0.10 | 4.1 | ±0.10 | 4.6 | ±0.07 | - | 3.03 | 0.21 | 93.22 |
| | Au3 | 18–30 | 5.5 | ±0.08 | 4.3 | ±0.08 | 4.6 | ±0.01 | 0.1 | 3.98 | 0.26 | 93.18 |
| | Cg1 | 30–45 | 5.9 | ±0.05 | 4.8 | ±0.07 | 5.5 | ±0.08 | - | 0.15 | 0.01 | 99.57 |
| | Cg2 | >45 | 6.0 | ±0.08 | 4.9 | ±0.05 | 5.6 | ±0.02 | - | 0.10 | 0.00 | 99.72 |
| P2 | M1 | 0–10 | 7.4 | ±0.08 | 6.9 | ±0.04 | 7.1 | ±0.01 | 20.9 | 15.89 | 1.30 | 67.20 |
| | M2 | 10–25 | 7.5 | ±0.08 | 7.1 | ±0.01 | 7.2 | ±0.02 | 26.9 | 13.88 | 1.15 | 69.32 |
| | M3 | 25–42 | 7.5 | ±0.10 | 7.0 | ±0.02 | 7.3 | ±0.03 | 28.6 | 14.82 | 1.14 | 70.97 |
| | Oa | 42–55 | 7.2 | ±0.07 | 6.7 | ±0.02 | 7.2 | ±0.04 | 4.3 | 18.05 | 1.28 | 62.78 |
| | Lcca | 55–70 | 6.8 | ±0.05 | 6.6 | ±0.10 | 6.9 | ±0.03 | 0.9 | 11.69 | 0.70 | 75.86 |
| | Cgg | >70 | 6.4 | ±0.08 | 6.5 | ±0.02 | 6.3 | ±0.03 | - | 0.54 | 0.03 | 98.11 |
| P3 | Oe | 0–14 | 6.1 | ±0.08 | 4.5 | ±0.27 | 5.3 | ±0.16 | 0.1 | 34.22 | 2.09 | 31.72 |
| | Oa1 | 14–34 | 5.4 | ±0.10 | 4.6 | ±0.02 | 4.8 | ±0.15 | 0.3 | 29.70 | 1.96 | 46.13 |
| | Oa2 | 34–62 | 5.8 | ±0.06 | 4.6 | ±0.02 | 5.0 | ±0.22 | 0.2 | 34.12 | 2.03 | 32.54 |
| | Oa3 | 62–70 | 5.9 | ±0.07 | 4.7 | ±0.04 | 5.2 | ±0.28 | 0.3 | 31.56 | 1.83 | 39.83 |
| | Oa4 | 70–83 | 5.8 | ±0.08 | 4.7 | ±0.03 | 5.0 | ±0.10 | 0.1 | 32.03 | 1.61 | 37.43 |
| | Lc | 83–106 | 5.8 | ±0.09 | 3.8 | ±0.31 | 5.0 | ±0.11 | - | 11.66 | 0.43 | 83.19 |
| | Cgg | 106–120 | 5.6 | ±0.13 | 3.9 | ±0,20 | 4.8 | ±0.11 | - | 2.20 | 0.09 | 95.51 |

### 3.2.3. pH Value and $CaCO_3$ Content

In soils P1 and P3, the measured pH in $H_2O$ does not exceed 6 (Table 2). Higher pH values are found in the lower part of the P1 profile with a slightly higher value also recorded in the surface horizon (Au1). This soil is strongly acidic in the upper part of the profile and acidic in the deeper mineral horizons (based on pH in KCl). The lowest pH values were obtained in KCl, the highest in $H_2O$, and measurements in $CaCl_2$ gave intermediate values, which is consistent with the ionic strength of the solutions used. In P3, the lowest pH values were obtained in $CaCl_2$, which were approximately equal in the organic horizons (around 4.5), while the pH significantly decreased in the Lc and Cgg horizons to a value lower than 4. The highest pH values were found in soil P2. The organic horizons have an alkaline reaction (pH 7.5 in $H_2O$) and change to slightly acidic in the lower, mineral part of the profile (pH 6.5 in $H_2O$). These results are strongly correlated with the chemical nature of the waters feeding this soil [44]. The alkaline values of the organic horizons in P2 are closely related to the $CaCO_3$ content in the moorsh horizons, which ranges from 21 to 28.8% and increases with depth in the profile (Table 3). In the peat level (Oa), the $CaCO_3$ content decreases significantly to 4%, and in deeper horizons, it does not represent a significant percentage of the soil. In the other soils, the $CaCO_3$ content is very low, and would not be expected to have any significant influence on soil properties.

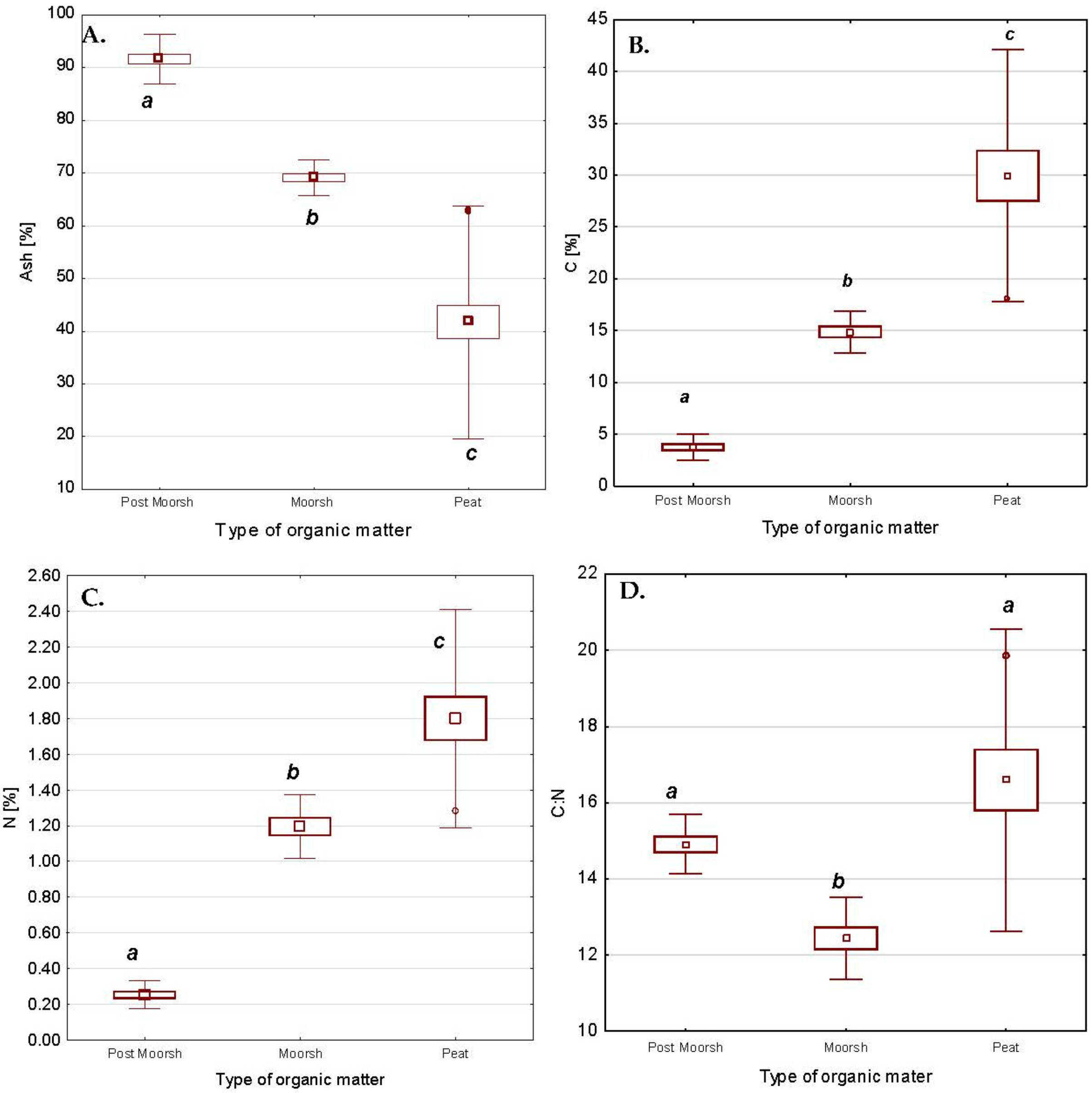


**Figure 3.** Variability range of the Ash, C, N content and C:N ratio in groups of organic horizons categorised by type of organic matter for Post Moorsh (A horizons of P1), Moorsh (M horizons of P2), and Peat (O horizons of P2 and P3). Explanation: central squares—mean values; boxes—mean values ± standard errors whiskers—mean values ± 2*standard deviations; circle—outliers; asterisk—extreme; letters a, b, c—significance of differences (homogeneous groups of means) by RIR Tukey test (at $p < 0.05$).

**Table 3.** Amorphous Fe and Al content and PSC.

| Soil Profile | Genetic Horizon | Depth (cm) | Feox | | | | Alox | | | | PSC [1] | | | |
|---|---|---|---|---|---|---|---|---|---|---|---|---|---|---|
| | | | (g kg$^{-1}$) | | | | | | | | | | | |
| 1 | Au1 | 0–8 | 16.73 | ± | 0.43 * | a | 1.03 | ± | 0.24 | a | 8.88 | ± | 0.34 | a |
| | Au2 | 8–18 | 20.60 | ± | 0.62 | a | 1.49 | ± | 0.17 | a | 11.04 | ± | 0.22 | a |
| | Au3 | 18–30 | 19.60 | ± | 0.18 | a | 1.35 | ± | 0.04 | a | 10.48 | ± | 0.07 | a |
| | Cg1 | 30–45 | 5.52 | ± | 0.06 | | 0.42 | ± | 0.03 | | 2.97 | ± | 0.02 | |
| | Cg2 | >45 | 0.53 | ± | 0.29 | | 0.12 | ± | 0.02 | | 0.33 | ± | 0.15 | |
| 2 | M1 | 0–10 | 31.75 | ± | 0.10 | a | 1.40 | ± | 0.03 | a | 16.58 | ± | 0.04 | a |
| | M2 | 10–25 | 32.89 | ± | 1.04 | a | 1.46 | ± | 0.07 | a | 17.17 | ± | 0.55 | a |
| | M3 | 25–42 | 34.52 | ± | 1.13 | a | 1.21 | ± | 0.05 | a | 17.86 | ± | 0.59 | a |
| | Oa | 42–55 | 45.05 | ± | 0.71 | b | 7.5 | ± | 0.14 | b | 26.15 | ± | 0.43 | b |
| | Lcca | 55–70 | 78.53 | ± | 7.71 | | 13.29 | ± | 0.29 | | 45.91 | ± | 4.00 | |
| | Cgg | >70 | 13.73 | ± | 0.45 | | 0.66 | ± | 0.10 | | 7.19 | ± | 0.27 | |
| 3 | Oe | 0–14 | 59.48 | ± | 3.39 | a | 2.50 | ± | 0.00 | a | 30.99 | ± | 1.70 | a |
| | Oa1 | 14–34 | 23.78 | ± | 1.14 | b | 3.46 | ± | 0.28 | a | 13.62 | ± | 0.71 | b |
| | Oa2 | 34–62 | 17.57 | ± | 0.14 | c | 2.01 | ± | 0.06 | ab | 9.79 | ± | 0.10 | c |
| | Oa3 | 62–70 | 17.70 | ± | 0.33 | c | 17.63 | ± | 0.81 | c | 17.66 | ± | 0.57 | d |
| | Oa4 | 70–83 | 13.47 | ± | 0.01 | c | 13.44 | ± | 0.19 | d | 13.45 | ± | 0.09 | b |
| | Lc | 83–106 | 15.93 | ± | 0.24 | | 7.11 | ± | 0.27 | | 11.52 | ± | 0.25 | |
| | Cgg | 106–120 | 9.12 | ± | 1.56 | | 1.82 | ± | 0.39 | | 5.47 | ± | 0.98 | |

Explanation: [1] PSC—P sorption capacity; * mean value ± SD; a, b, c, d—groups of homogeneous means at $p = 0.05$ (values marked with the same letter are not statistically significantly different).

### 3.2.4. Content of C and N

The content of total organic carbon (C) varies strongly in the investigated soils (Table 2). P1 contains 4.0–4.3% C in the humus Au horizons; in P2 this value increases to 14–16% in the moorsh horizons and up to 18% in the Oe horizon, before decreasing to less than 12% in the Lcca horizon (which contains ferruginous gyttja and carbonate skeletons) due to a significant proportion of mineral admixtures. Soil P3 (peat soil) is characterised by a C value between 30 and 34% and shows a few percent variation between organic horizons, reflecting variable conditions during deposition of organic layers, as well as differences in peat species compositions. In the sub-bedding layers, the content of C decreases significantly reaching 2.2% directly beneath the organic horizon, which is influenced by the admixtures of mossy peat occurring alongside gyttja and sandy material in this horizon (Cgg). The N content is strongly related to OM transformations. The lowest N values occur in P1 and the highest values in P3 (Table 2). Changes in C and N content highlight the transformations occurring in the studied soils, with values decreasing in the order P3, P2, P1, i.e., inverse to the ash content of the soils, thus confirming the organic matter transformation taking place in the soils (Figure 3B,C).

### 3.2.5. C:N Ratio

The C:N ratio is an important indicator for assessing OM transformation in soils and is, therefore, also a commonly used indicator of the rate of peat decomposition. This approach is based on the residual enrichment of N relative to C during OM mineralisation, meaning the lower the C:N ratio, the more decomposed the peat material [63,64]. The recorded C:N ratio values in this study follow this anticipated trend (Table 4). In the studied organic horizons, the C:N ratio takes values in the range of 12–20, whereas in mineral horizons the C:N ratio range is markedly wider. The C:N ratio in P1 is very homogeneous, ranging between values of 14.5 and 15.3. Its value is higher in the surface horizon Au1, which can be related to fresh organic matter input from the soil surface. In soil P2, the C:N ratio range is slightly narrower than in P1; its values in the organic horizons are 12–14 and in the lower horizons they reach almost 20. The highest values of the C:N ratio (15–20) are recorded in the organic layers in P3. These values increase with increasing depth in the profile, reaching

a maximum C:N value of 27 in the Lc horizon. In the P3 profile, the organic matter is the least decomposed, and, hence, has the highest C:N values. In the profile distribution of the peat soil, the C:N ratios appear to reflect the process of peat accumulation under reducing conditions (Oa1–Oa4 horizons), and therefore show increasing values with depth in these layers. The surface horizon (Oe) has a slightly different character. The upper peat layer at depths of around 0–30 cm was excavated in this area [43]; the material present in Oe is, therefore, the youngest and remains in the accumulation phase. This layer is characterised by fresh organic material with a lower degree of decomposition, which accounts for why a larger C:N ratio value is observed here than in the lower organic horizons.

**Table 4.** Indicators of P behaviour in the soils under the OM transformation.

| Soil Profile | Genetic Horizon | Depth (cm) | PSD [1] (%) | | $I_P$ (%) | | C:N | | C:Pt | | N:Pt | | C:N:Pt |
|---|---|---|---|---|---|---|---|---|---|---|---|---|---|
| 1 | Au1 | 0–8 | 19.1 ± 0.4 | a | 1.03 | a | 15:1 | a | 16:1 | a | 1:1 | a | 16:1:1 |
| | Au2 | 8–18 | 15.4 ± 0.2 | b | 0.48 | b | 14:1 | b | 12:1 | a | 1:1 | a | 14:1:1 |
| | Au3 | 18–30 | 14.2 ± 0.2 | b | 0.41 | b | 15:1 | a | 18:1 | a | 1:1 | a | 18:1:1 |
| | Cg1 | 30–45 | 7.0 ± 1.7 | | 0.35 | | 15:1 | | 3:1 | | 1:1 | | 15:1:5 |
| | Cg2 | >45 | 9.6 ± 5.2 | | 0.58 | | - | | 6:1 | | - | | - |
| 2 | M1 | 0–10 | 11.0 ± 0.4 | a | 0.28 | a | 12:1 | a | 54:1 | a | 4:1 | a | 54:4:1 |
| | M2 | 10–25 | 10.7 ± 0.6 | a | 0.23 | a | 12:1 | a | 46:1 | a | 4:1 | a | 46:4:1 |
| | M3 | 25–42 | 11.0 ± 0.5 | a | 0.21 | a | 13:1 | b | 49:1 | a | 4:1 | a | 49:4:1 |
| | Oa | 42–55 | 8.9 ± 0.3 | b | 0.15 | a | 14:1 | c | 57:1 | b | 4:1 | b | 57:4:1 |
| | Lcca | 55–70 | 1.7 ± 0.2 | | 0.08 | | 17:1 | | 92:1 | | 6:1 | | 92:6:1 |
| | Cgg | >70 | 1.1 ± 0.1 | | - | | 17:1 | | 34:1 | | 2:1 | | 34:2:1 |
| 3 | Oe | 0–14 | 4.2 ± 0.3 | a | 0.14 | a | 16:1 | a | 163:1 | a | 10:1 | a | 163:10:1 |
| | Oa1 | 14–34 | 4.1 ± 0.3 | a | 0.09 | a | 15:1 | b | 244:1 | b | 16:1 | b | 244:16:1 |
| | Oa2 | 34–62 | 4.4 ± 0.8 | a | 0.10 | a | 17:1 | c | 338:1 | c | 20:1 | c | 338:20:1 |
| | Oa3 | 62–70 | 3.1 ± 0.1 | a | 0.06 | a | 17:1 | d | 287:1 | d | 17:1 | d | 288:17:1 |
| | Oa4 | 70–83 | 3.3 ± 0.7 | a | - | | 20:1 | e | 356:1 | e | 18:1 | e | 356:18:1 |
| | Lc | 83–106 | 3.3 ± 0.7 | | - | | 27:1 | | 194:1 | | 7:1 | | 194:7:1 |
| | Cgg | 106–120 | 2.4 ± 0.1 | | - | | 24:1 | | 81:1 | | 3:1 | | 81:3:1 |

[1] PSD—P saturation degree; Ip—solubility index of P; Pt—total P; a, b, c, d—groups of homogeneous means at $p = 0.05$ (values marked with the same letter are not statistically significantly different.

In P2, the M1–M3 moorsh horizons were formed as a result of peat mineralisation due to lowering of the water level the exposure of these horizons to the air, changing the reducing conditions to aerobic ones. This resulted in a rapid acceleration of organic matter decomposition and intensive mineralisation of its components, accompanied by the release of N from organic compound structures. This process is evidenced by the increasing abundance of bacterial microflora actively participating in denitrification processes, as confirmed by numerous previous studies [65–67]. Hence, in the M1-M2 moorsh horizons the C:N ratio value, equal to 12, is the smallest. Its slight increase in the M3 (C:N = 13) and Oa (C:N = 14) horizons relates to the oxidoreductive conditions changing with depth and to the weaker rate of OM transformations.

Our results, and the relationships between them, are consistent with previous studies of secondary transformations in organic soils [17,68,69]. The higher ratio values in the surface layer at P1 relative to P2 are related to the influx of fresh plant material, which disrupts the processes observed at P2; at P3, the C:N ratio increased significantly. The comparison in C:N ratio between the soils in groups of organic horizons is shown in Figure 3D. Significant differences were confirmed in P2 (murshic soil) in comparison to P1 and P3 (Figure 3D).

### 3.2.6. Sorption Properties

The sorption properties of the investigated soils also vary considerably; the results of these measurements are shown in Table S2. The weakest sorption properties are observed at P1. The CEC sorption capacity in the surface horizons of P1 ranged from 16 to 28.8 $cmol_c$ $kg^{-1}$, with the highest value near to the soil surface, while in the mineral horizons it decreased considerably to 1.6–2.5 $cmol_c$ $kg^{-1}$. A significant proportion of the CEC is from exchangeable base cations (67–80% in the organic horizons); the highest proportion occurs in the surface horizon, where exchangeable $Ca^{2+}$ (17.4 $cmol_c$ $kg^{-1}$) dominates. Hydrolytic acidity is between 5 and 5.7 $cmol_c$ $kg^{-1}$ in the organic horizons and decreases with increasing profile depth. The CEC values are much higher in P2 and P3 soils. In P2, values from 101.5 to 137.7 $cmol_c$ $kg^{-1}$ are recorded in the organic horizons, while in P3 the lowest CEC value is observed near the surface in the Oe horizon (83 $cmol_c$ $kg^{-1}$). In the deeper organic layers, the CEC increases considerably with significant variability, with the highest value of 257 $cmol_c$ $kg^{-1}$ recorded in the middle organic horizon (Oa2). The main contribution to the sorption properties is the $Ca^{2+}$ cation (Table S2). The large variation in sorption properties within the P3 profile is probably related to the variable parameters of the water supplying this area. HA values are lowest in P2 and highest in P3 where they reach up to 30.6 $cmol_c$ $kg^{-1}$ in the organic horizons (Oa1).

### 3.2.7. Alox and Feox Content

The Al and Fe forms extracted into solution by the oxalate method are mainly active, non-crystalline forms. These are important components in determining P retention in the soil, especially in non-carbonate, acidic soils [23,26].

The highest Feox contents were observed in P2 (Table 3), where values exceed 30 g $kg^{-1}$ in horizons M1-M3 and rise to almost 80 g $kg^{-1}$ in horizon Lcca; below this horizon, Feox content decreases significantly. In P3, high Feox content (60 g $kg^{-1}$) is observed in the surface horizon, Oe. This may relate to the fact that this horizon remains in the zone of frequent water level fluctuations, and, therefore, more intense redox reactions occur, resulting in the accumulation of Fe in the soil. In deeper organic horizons, lower Feox values are recorded. In the P1 soil, Feox content in the Au horizons is approximately equal, reaching a maximum of 20 g $kg^{-1}$, whereas in the lower horizons of the profile its abundance decreases significantly.

The Alox content of the upper organic horizons of the investigated soils is relatively consistent between sites (Table 3). The Alox content increases in the upper horizons at these sites in the order: P1 < P2 < P3. However, the investigated soils show considerable differences in Alox content in the lower parts of the profiles. In soil P1, Alox content decreased significantly in the mineral Cg horizons to a value of <0.42 g $kg^{-1}$, whereas, in P2 and P3 much higher Alox values were observed in the deeper organic horizons. The highest values of Alox, are equal to 13 and 18 g $kg^{-1}$, which occur in the Lcca horizon of P2 and the Oa3 horizon of P3, respectively.

### 3.2.8. The Phosphorus Sorption Capacity

The P sorption capacity to (PSC) is an estimated value based on the Alox and Feox content of the investigated soils (Table 3). The summed amounts of Alox and Feox are widely used to estimate the amount of P that can be retained in the soil and PSC is also an important factor in regulating the release of P from soil to water [23,70].

The highest PSC was found in organic soils (sites P2 and P3). The highest value of PSC was observed in the surface horizon (Oe) in P3, where it was more than 30 g $kg^{-1}$ and decreased in the deeper organic horizons by more than half. In P2, the value of PSC was about 15 g $kg^{-1}$ in the moorsh horizons M1-M3 and increased to a value of about 45 g $kg^{-1}$ in the Lcca horizon, which has significantly elevated Fe and Al content. In the organic horizons of the post-murshic soil (P1), PSC had the lowest values compared to the other profiles (about 10 g $kg^{-1}$) and decreased significantly in the deeper mineral horizons of the profile.

### 3.3. Phosphorus Behaviour in Soil

#### 3.3.1. Total and Mineral Phosphorus

In the surface horizons of the studied soils, in which the highest accumulation of P usually occurs, the Pt content reaches 2–3 g $kg^{-1}$ (Table 5); the lowest values occur in P3 (peat soil) and the highest values are recorded in P2 (murshic soil). The significance of differences between mean results in the organic horizons of each profile was demonstrated by the Tukey test, the results of which are presented in Table 4. These results are comparable to previous studies on organic soils [17]. The high Pt values in P2 may be significantly influenced by high $CaCO_3$ content. $CaCO_3$ is highly active in P binding by forming chemically insoluble compounds or by acting as a cation forming bridging links between P and OM, for example; these links may increase P retention in the soil. Link formation may also be promoted by the high pH of the P2 soil. Analysis of the r-Pearson correlation coefficients in the organic horizons of the studied soils confirms the statistical significance of the positive relationship between Pt, pH, and $CaCO_3$ (Table 6).

In terms of the distributions of the phosphorous forms, in the mineral horizons in the lower part of the soil profiles, the Pt content decreases significantly (Table 4). A similar distribution is recorded in terms of Pox and Pmin (i.e., P bound by the mineral fraction of soils) content. The lowest Pox values in organic horizons occur in P3 (1.0–1.3 g $kg^{-1}$) and the highest values are recorded in P2 (1.8–2.3 g $kg^{-1}$), an effect which is likely related to the aforementioned role of Ca and pH in P binding. In P1 Pox content is intermediate in the Au horizons, ranging from 1.48 to 1.7 g $kg^{-1}$. The Pmin content follows a very similar trend to Pox in all horizons, however, its values tend to be lower (Table 4). The extraction process of Pox using an oxalate mixture (organic) may account for this difference. Additionally, compounds connected with active forms of Al and Fe probably pass into the extraction solution, potentially in addition to those containing Al and Fe in their composition [71].

**Table 5.** Phosphorus forms in soils under study.

| Soil Profile | Genetic Horizon | Depth (cm) | $P_W$ [1] | | $P_{CaCl2}$ | | $P_{M3}$ | | Porg | | Pmin | | Pox | | Pt | |
|---|---|---|---|---|---|---|---|---|---|---|---|---|---|---|---|---|
| | | | (mg kg$^{-1}$) | | | | | | | | | | | | | |
| 1 | Au1 | 0–8 | 27.9 ± 4.0 * | a | 15.3 ± 1.8 | a | 70.4 ± 14.1 | a | 1522.5 ± 91.9 | a | 1193.5 ± 106.1 | a | 1692.8 ± 95.8 | a | 2716.0 ± 14.1 | a |
| | Au2 | 8–18 | 11.8 ± 0.1 | b | 3.2 ± 0.2 | b | 50.1 ± 10.0 | b | 1173.0 ± 7.1 | ab | 1277.5 ± 5.7 | a | 1702.8 ± 7.4 | a | 2450.5 ± 1.4 | ab |
| | Au3 | 18–30 | 8.9 ± 0.6 | b | 2.2 ± 0.1 | b | 42.3 ± 8.5 | c | 1032.0 ± 103.9 | b | 1155.5 ± 2.8 | a | 1483.3 ± 13.1 | a | 2187.5 ± 101.1 | b |
| | Cg1 | 30–45 | 1.6 ± 0.2 | | <0.32 ** | | 6.2 ± 1.2 | | 86.5 ± 21.7 | | 357.4 ± 7.2 | | 206.9 ± 47.8 | | 443.9 ± 14.5 | |
| | Cg2 | >45 | 0.8 ± 0.1 | | <0.32 | | 3.5 ± 0.7 | | 49.4 ± 13.5 | | 97.3 ± 1.9 | | 27.3 ± 1.9 | | 146.8 ± 15.4 | |
| 2 | M1 | 0–10 | 8.1 ± 0.5 | a | 5.7 ± 1.4 | a | 9.1 ± 1.9 | a | 1323.5 ± 181.0 | a | 1599.3 ± 28.6 | a | 1825.5 ± 67.2 | a | 2922.8 ± 152.4 | a |
| | M2 | 10–25 | 6.9 ± 0.2 | a | 2.9 ± 1.2 | ab | 5.8 ± 1.2 | b | 1376.5 ± 105.4 | a | 1651.0 ± 32.5 | a | 1838.8 ± 51.3 | a | 3027.5 ± 72.8 | a |
| | M3 | 25–42 | 6.5 ± 0.1 | a | 2.5 ± 1.2 | ab | 5.6 ± 1.1 | b | 1317.5 ± 34.6 | a | 1694.3 ± 3.9 | a | 1960.5 ± 161.9 | a | 3011.8 ± 30.8 | a |
| | Oa | 42–55 | 4.7 ± 1.0 | a | 1.0 ± 0.1 | b | 5.5 ± 1.1 | b | 1211.8 ± 90.9 | a | 1929.5 ± 19.1 | b | 2315.5 ± 34.6 | b | 3141.3 ± 71.8 | a |
| | Lcca | 55–70 | 1.0 ± 0.0 | | <0.32 | | 0.8 ± 0.2 | | 619.0 ± 40.3 | | 648.5 ± 6.4 | | 765.0 ± 7.1 | | 1267.5 ± 33.9 | |
| | Cgg | >70 | <0.48 ** | | <0.32 | | 1.2 ± 0.2 | | 73.4 ± 24.6 | | 85.1 ± 1.3 | | 80.8 ± 1.3 | | 158.5 ± 23.3 | |
| 3 | Oe | 0–14 | 2.9 ± 0.2 | a | 1.7 ± 0.1 | a | 8.5 ± 0.7 | a | 1048.0 ± 184.6 | a | 1051.3 ± 27.2 | a | 1308.8 ± 30.1 | a | 2099.3 ± 157.3 | a |
| | Oa1 | 14–34 | 1.1 ± 0.0 | a | 0.9 ± 0.0 | a | 5.9 ± 0.3 | b | 990.3 ± 112.5 | a | 226.0 ± 6.1 | b | 551.0 ± 14.1 | b | 1216.3 ± 106.4 | b |
| | Oa2 | 34–62 | 1.0 ± 0.0 | a | 0.7 ± 0.0 | a | 3.7 ± 0.2 | c | 758.3 ± 123.4 | a | 251.0 ± 1.4 | b | 430.8 ± 86.3 | b | 1009.3 ± 124.8 | b |
| | Oa3 | 62–70 | 0.6 ± 0.2 | a | <0.32 | a | 1.5 ± 0.2 | d | 682.2 ± 138.5 | a | 414.1 ± 16.7 | bc | 539.0 ± 7.1 | b | 1096.3 ± 155.2 | b |
| | Oa4 | 70–83 | <0.48 | a | <0.32 | a | 1.2 ± 0.3 | d | 658.9 ± 17.6 | a | 315.6 ± 1.4 | b | 450.4 ± 90.7 | b | 974.5 ± 19.1 | b |
| | Lc | 83–106 | <0.48 | | <0.32 | | 1.1 ± 1.2 | | 262.4 ± 90.5 | | 338.1 ± 7.0 | | 380.3 ± 88.7 | | 600.5 ± 97.6 | |
| | Cgg | 106–120 | <0.48 | | <0.32 | | 1.4 ± 1.7 | | 142.4 ± 17.4 | | 128.0 ± 11.0 | | 133.5 ± 20.9 | | 270.4 ± 28.3 | |

[1] Pw—water-extractable P forms; $P_{CaCl2}$—easily soluble P extracted in 0.01 mol/L $CaCl_2$ solution; $P_{M3}$—available P by the Mehlich III method; Porg—organic P; Pmin—mineral P; Pox—soil oxalate P; Pt—total P. * mean value ± SD. ** below the detection level; a, b, c, d—groups of homogeneous means at $p = 0.05$ (values marked with the same letter are not statistically significantly different.

**Table 6.** Correlation coefficients for selected basic soil properties and calculated parameters, and P forms in organic horizons of soils under study. Correlation coefficients significant at $p < 0.05$ are shown in bold.

| Variable | $P_W$ [1] | $P_{CaCl2}$ | $P_{M3}$ | Porg | Pmin | $P_{ox}$ | Pt |
|---|---|---|---|---|---|---|---|
| $pH_{H2O}$ | −0.10 | 0.03 | −0.42 | 0.55 | **0.76** | **0.63** | **0.73** |
| $pH_{KCl}$ | −0.20 | −0.03 | −0.55 | 0.49 | **0.64** | 0.50 | **0.62** |
| $pH_{CaCl2}$ | 0.00 | 0.10 | −0.32 | **0.61** | **0.79** | **0.69** | **0.78** |
| $CaCO_3$ | 0.00 | 0.12 | −0.30 | 0.56 | **0.62** | 0.49 | **0.64** |
| $Fe_{ox}$ | −0.20 | −0.14 | −0.31 | 0.39 | 0.51 | 0.45 | 0.50 |
| $Al_{ox}$ | −0.44 | −0.43 | −0.41 | **−0.65** | −0.45 | −0.51 | −0.54 |
| Ash | **0.73** | 0.57 | **0.77** | 0.55 | **0.61** | **0.70** | **0.62** |
| C | **−0.75** | **−0.60** | **−0.77** | **−0.59** | **−0.65** | **−0.73** | **−0.66** |
| N | **−0.76** | −0.57 | **−0.85** | −0.42 | −0.50 | **−0.61** | −0.51 |
| C:N | −0.32 | −0.36 | −0.13 | **−0.83** | **−0.79** | **−0.77** | **−0.85** |
| N:P | **−0.68** | −0.57 | **−0.64** | **−0.75** | **−0.84** | **−0.90** | **−0.86** |
| C:Pt | **−0.64** | −0.55 | **−0.58** | **−0.80** | **−0.86** | **−0.92** | **−0.89** |
| C:$P_{org}$ | **−0.68** | **−0.58** | **−0.65** | **−0.78** | **−0.77** | **−0.85** | **−0.82** |
| Ip | **0.99** | **0.92** | **0.93** | **0.58** | 0.27 | 0.44 | 0.39 |
| PSD | **0.89** | **0.78** | **0.87** | **0.65** | 0.57 | **0.69** | **0.63** |

[1] Pw—water-extractable P forms; $P_{CaCl2}$—easily soluble P extracted using 0.01 mol/L CaCl2 solution; $P_{M3}$—available P by the Mehlich III method; Porg—organic P; Pmin—mineral P; Pox—soil oxalate P; Pt—total P; Ip—solubility index of P; PSD—P saturation degree.

### 3.3.2. Organic Phosphorus

In the surface horizons, Porg values range between 1.0 and 1.5 g $kg^{-1}$ (Table 4) and its content follows the series P1 > P2 > P3; Porg content, therefore, increases with decreasing soil moisture content and is inversely proportional to C in these soils. However, the value of Porg increases in the horizons influenced by anaerobic conditions (P3) (Table S3). Within horizons in aerobic conditions, the organic phosphorous proportion decreases with increasing degree of drainage (Table S3); this effect is clearly visible vertically in soil profiles (Figure 4) and the comparison between profiles (Figure 5). In the peat soil (P3), Porg accounts for 65–80% of Pt in the organic horizons in reducing conditions (Oa1-Oa4 horizons) and nearly 50% of Pt in the surface horizon (Oe), where we observed frequent fluctuations of the water level. In P2 the percentage of Porg decreased to about 45% in all moorsh horizons (M1–M3), whereas in the underlying Oa horizon the value was <40%. Similar values were obtained for the humus horizons in P1 (Table S3). The higher proportion of Porg in the organic horizons in peat soil (P3) indicates P accumulation has taken place, which is related to the presence of P in plant tissues in the form of organic molecules; this finding is also confirmed by other authors' results (e.g., [17]). Under the influence of changing humidity conditions and access to oxygen (Oe horizons in P3, M horizons in P2 and Au horizons in P1), decomposition and mineralisation of OM significantly increase in intensity. This phenomenon is accompanied by increased microbiological activity which accelerates the release of P from organic structures, increasing the proportion of mineral forms of P (e.g., Pmin). In addition to the mineralisation process, humification of OM may also occur, leading to qualitative remodelling of humic substances in the organic horizons. Mineral forms of P may also be used by organisms, which would lead to a decrease in Pmin in surface horizons. This pattern is recorded in the Au1 horizon in P1 where an increase in the percentage of Porg and a decrease in Pmin are observed (Table S3). The Porg content in the organic horizons was confirmed to be positively correlated with pH measured in $CaCl_2$, however, this relationship was not statistically confirmed for pH values in $H_2O$ and KCl (Table 6). In terms of other factors, organic carbon (C) and Alox content are both correlated with Porg (Table 6). The correlation between Alox and Porg, however, appears to be more closely related to Alox content varying due to habitat variability rather than the direct effect of Alox on Porg transformations. However, the relationship between C and Porg differs from the Alox/Porg relationship; this difference may be explained by the fact

that progressive mineralisation in organic horizons tends to favour the release of Porg as C decreases.

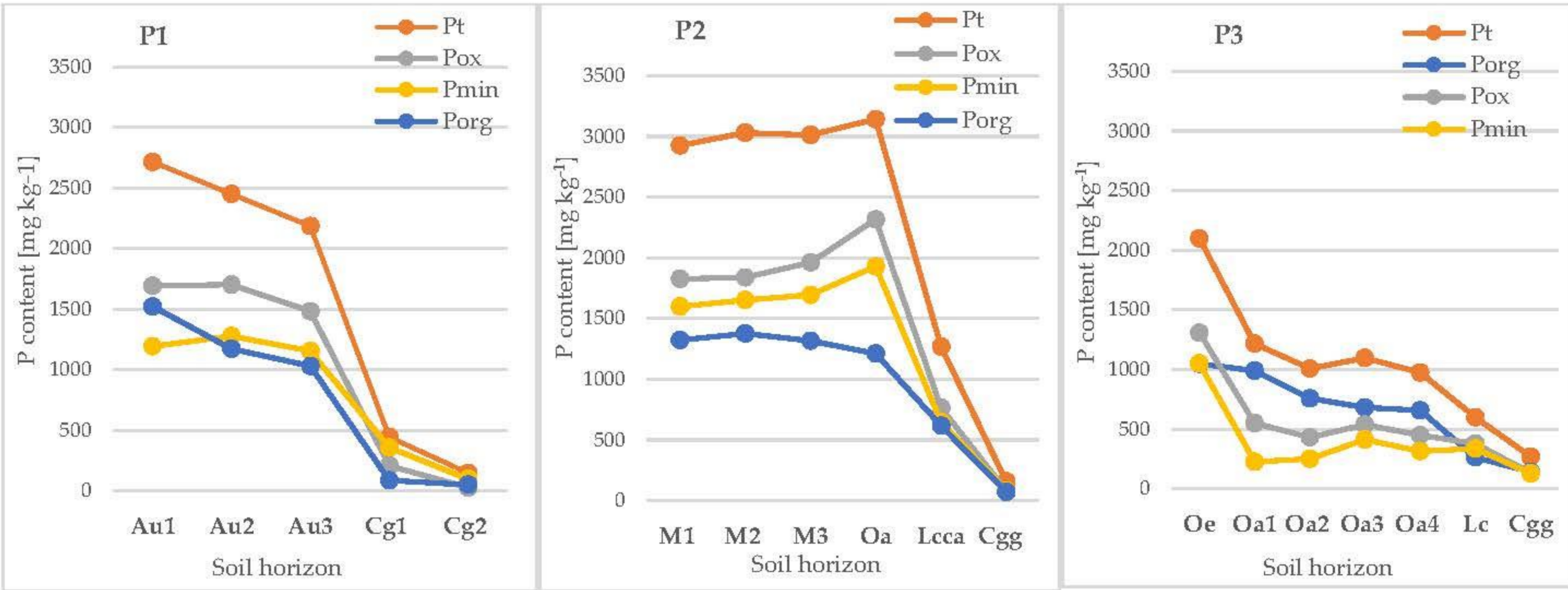


[1]Porg - organic P; Pmin - mineral P; Pt - total P

**Figure 4.** Quantitative relationships between P forms in the studied soils: Pt[1], Pox, Pmin, Porg; Pox—soil oxalate P.

The content of both mineral (Pox, Pmin) and organic forms of P (Porg) and, consequently, the Pt content value is strongly related to the effects of changing air–water conditions in the studied soils but also to each soil's basic properties. This is well-illustrated by the example of soil P2, where higher values of the aforementioned P forms are associated with higher pH and $CaCO_3$ content in this profile (Figure 4); aerobic conditions are also recorded in the moorsh horizons, with significantly higher Feox content in this profile.

### 3.3.3. Soluble Forms of Phosphorus

The content of soluble P forms is of crucial importance for the quality of soils, as well as for the quality of the nearby surrounding ecosystem. These P forms were analysed using three methods (Table 4): (i) available P forms ($P_{M3}$), which is of fundamental importance from an agricultural point of view, determined according to the Mehlich III method; (ii) extraction in 0.01 mol/L $CaCl_2$ solution ($P_{CaCl2}$), which gives information about the content of mobile P forms that are easily leached into solution under chemically similar conditions to the ionic strength of the soil solution; (iii) extraction in distilled water (Pw), which allows the content of highly water-soluble P forms to be assessed. These forms of P are very easily leached into the environment, e.g., during rainfall, and their presence at increased levels may pose a threat due to the potential for superficial or deep leaching of P from the soil profile. This P may then enter the surrounding environment causing hazards such as eutrophication of surrounding watercourses and reservoirs (e.g., [72,73]).

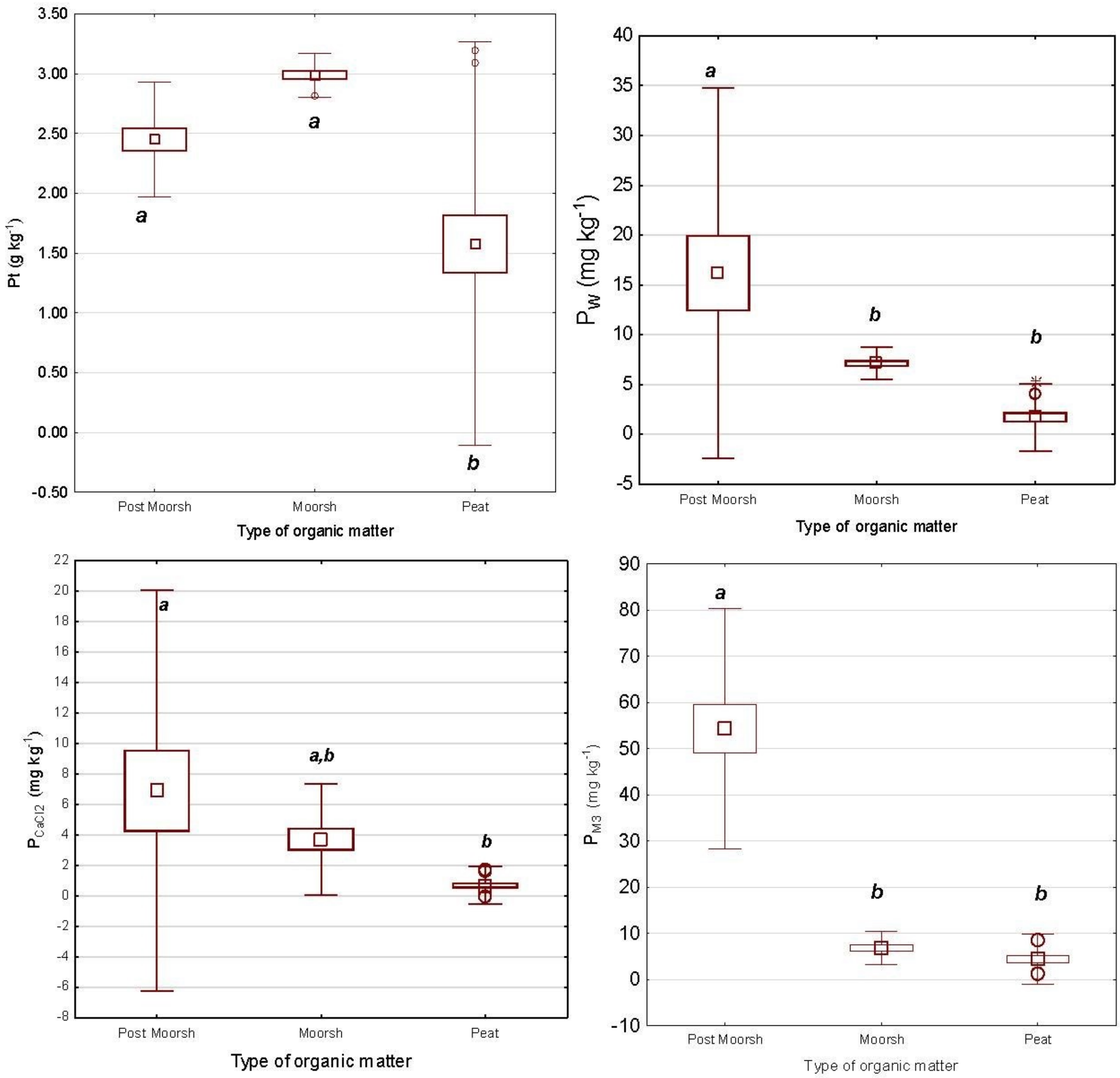


**Figure 5.** Comparison of Pt and soluble P forms (Pw, $P_{CaCl2}$, $P_{M3}$)[1] in groups of organic horizons depending on the degree of OM transformation: Peat, Moorsh, Post Moorsh horizons. Explanation: central squares–mean values; boxes–mean values ± standard errors; whiskers—mean values ± 2*standard deviations; circle—outliers; asterisk—extreme; letters a, b, c—significance of differences (homogeneous groups of means) by RIR Tukey test (at $p < 0.05$). [1] Pt—total P; Pw—water-extractable P forms; PCaCl2—easily soluble P extracted in 0.01 mol/L CaCl2 solution; PM3—available P by the Mehlich III method.

The highest $P_{M3}$ content values in the studied soils are found in the surface horizons of the studied soils; their $P_{M3}$ content then decreases with the depth in the soil profiles. The highest $P_{M3}$ values were observed in the postmurshic soil (P1), where the $P_{M3}$ content in the Au humus horizons ranges from 70.4 mg $kg^{-1}$ at the surface to 42.3 mg $kg^{-1}$ at the base of these horizons; these values are over 10 times higher than those recorded in the bedrock horizons Cg1 and Cg2 (Table 4). The significance of these changes inside the profile was verified by Tukey's post-hoc test, the results of which are presented in Table 4. The $P_{M3}$ content is nearly 10 times lower in the surface horizons of the murshic (P2) and peat (P3) soils than the value recorded in P1. In both P2 and P3 soils the content of $P_{M3}$ is very similar with values of around 9 mg $kg^{-1}$ in the organic surface horizons (M1 and Oe) and about 6 mg $kg^{-1}$ in deeper horizons, reaching a minimum of approximately 1 mg $kg^{-1}$ in the mineral horizons of these soils (Lcca, Lc, and Cgg). The amount of $P_{M3}$ represents

about 2–2.6% of Pt content in the postmurshic soil, which in murshic and peat soils has a smaller percentage, about 0.2–0.4% of Pt (Table S3).

The mobile P forms ($P_{CaCl2}$) (Table 4, Table S3) also have the highest values in the surface horizons with a gradual decrease recorded in the deeper organic horizons, reaching a minimum in the mineral horizons at the base of the soil profiles. The highest surface value of $P_{CaCl2}$ (15.3 mg kg$^{-1}$) was found in the postmurshic soil (P1), with a lower value of ~6 mg kg$^{-1}$ in the uppermost part of the murshic soil (P2), and the lowest value was recorded in the surface layer of the peat soil (P3) (2 mg kg$^{-1}$). The overall percentage of this component as a fraction of Pt is very small but follows the same trend: P1 (0.5% of Pt in Au1) > P2 (0.19% in M1) > P3 (0.08% in Oe). As in the case of other parameters, the $P_{CaCl2}$ content increases with increasing degrees of soil drainage, decomposition, and mineralisation of organic matter.

Along with the transformation of organic matter in the studied soils, a clear increase in surface Pw content was recorded from site P3 (about 3 mg kg$^{-1}$) to P2 (8 mg kg$^{-1}$) to P1 (30 mg kg$^{-1}$ in Au horizons) (Table 4). Analogous to the other P forms, the Pw content decreased significantly with depth in the profiles reaching very low values at the base of each profile, in some cases below the detection threshold (e.g., Cgg horizons in P2 and P3). The statistical significance of the differences between organic horizons in individual profiles was confirmed by Tukey's test (Table 4). A significantly higher Pw content was recorded in comparison to the $P_{CaCl2}$ values in the studied soils. This was an expected outcome as such relationships (Pw > $P_{CaCl2}$) are common, as evidenced by numerous reports from previous studies [74–77]. Higher P concentrations in deionised or distilled water extracts are associated with lower ionic strengths and the effect of different soil solution compositions on P solubility [78,79]. When using an extract with low ionic strength, there is also a higher probability of the dispersion of clay colloids [80,81], which further increases the Pw values. The easier transition of P into the water phase (Pw) relative to the weak salt solution phase ($P_{CaCl2}$) may represent a potential environmental problem. In particular, the consequences of intensive OM mineralisation in highly humid ecosystems must be considered, especially those adjacent to agricultural areas, where P availability to plants is essential, but an excess of its soluble form may pose a threat to surrounding watercourses [8,23,26].

Based on the results presented in this study, the content of soluble P forms (available and mobile P) in the studied soils is closely related to both the degree of environmental humidity and the intensification of OM transformations. This is evident in both the vertical distribution of values within each profile (Table 4) and laterally in terms of the groups of organic horizons by type (Figure 5), a result that was confirmed statistically. In the peat soil (P3), where reducing conditions dominate, soluble forms of P ($P_{M3}$, $P_{CaCl2}$, Pw) are present in the lowest concentrations compared to P1 and P2. At P3, P is only present as a component of undecomposed organic matter or may be strongly immobilised by OM; it may also remain in mineral combinations by mechanisms such as P fixing or complexing with components such as Fe, Al or Ca. The surface horizon of this soil (Oe), influenced by strong fluctuations in water level, is characterised by a higher degree of organic matter decomposition, accompanied by an increase in the content of mobile forms of P in this zone, as well as in the underlying zone, where the consequences of the change in oxidoreductive conditions are also seen. In the murshic soil (P2), occurring under reduced water level conditions after drainage, the mursh processes are already significantly advanced. The periodic aeration of organic material has influenced the onset of OM mineralisation and its accompanying gradual release of nutrients, hence, the increased content of mobile forms of P present. All measured P forms (available and mobile) are present in higher contents in this soil (P2) than in the peat soil (P3). The smallest fluctuations were observed for $P_{M3}$. The high pH and the $CaCO_3$ content in P2, which are highly likely to increase P retention in the soil, may be responsible for this observation, but these factors are considered more likely to fix P in its insoluble form. Of particular note is the very high content of Pw (close to $P_{M3}$ values), which is much higher than the $P_{CaCl2}$ in P2 (Figure S1).

In the postmurshic soil (P1) the superficial organic horizons are mainly influenced by aerobic conditions with reducing conditions only occasionally present in much of the profile. In this soil, the transformation of organic matter is highly advanced, as evidenced by the basic properties of this soil, e.g., C and Ash content (Table 2, Figure 3). In this soil, the measured values of available and mobile P forms reach their highest values at the surface with a profile distribution typical for P, with a decrease in P content with increasing profile depth. The results of grouping objects and properties (Pw, $P_{CaCl2}$, $P_{M3}$) clearly show the highest intensity of P solubility in the organic horizons of this soil (Figure S1). The higher solubility of P as an effect of dewatering of organic soil was observed earlier in the studies of the moorsh horizons and murshic soils (e.g., [17,24,25]) but the evidence of this effect within postmurshic soil, as a potentially arable soil, requires further attention. These results may have problematic implications for environmental quality as the intense P release into soil solution can lead to dispersion of P into the environment.

### *3.4. Indicators of Phosphorus Accumulation and/or Release in the Soil*

#### 3.4.1. Phosphorus Saturation Degree (PSD)

The P saturation degree (PSD) characterises the P reserves in the soil. The PSD method [53] estimates the proportion of the PSC filled with P, indicates the potential desorbability of soil P and is commonly used as a parameter for predicting water-soluble P [23,76,82–85]. The lowest PSD values (Table 5) were recorded in peat soil (P3), where a highly consistent PSD value of ~4% was observed in the upper peat horizons; slightly lower values were found in the deeper part of this profile. In the murshic soil (P2) PSD increases to around 10% in the organic horizons. A greater increase in PSD is observed in the organic horizons of the post murshic soil (P1), where it reaches nearly 20% in the surface horizon, ca. 15% in deeper horizons and 6–7% in the deepest part of the profile. A comparison of PSD variability in organic horizons is presented in Figure 6. The results show that the PSD of the studied soils increases according to the relationship P3 < P2 < P1, i.e., an increase with the degree of land drainage. This clearly indicates the direction of P transformations in accordance with the mineralisation degree of organic matter. Large PSD differences between profiles in the surface horizons indicate a high intensity of P release from transformed OM and organic molecules and simultaneous occupation by P of the available sorption sites in the soil. The PSD values decrease with depth in the soil profiles (Table 5), which confirms the important role of OM (or connections with OM) in the retention of released P in soil [23,26].

The PSD and PSC methods are not typically used in organic soils due to the previously interpreted major role of Ca, rather than Al and Fe, in P fixation. Nonetheless, we decided to use these parameters because they provide the opportunity to investigate the transformation of P in relation to Al and Fe in the soil profiles in this study. These indicators have previously been used in similar scenarios (e.g., [26]). In the studied soils, Ca plays a highly important role in soil sorption complexes, as presented earlier in this paper (Table S2). The Ca content is very high in P2, which is also reflected by the high pH value of this soil. The PSC and PSD results must, therefore, be interpreted with caution. However, the results presented earlier allow us to conclude that Al and Fe are increasingly important in P binding during OM transformation in the studied soils.

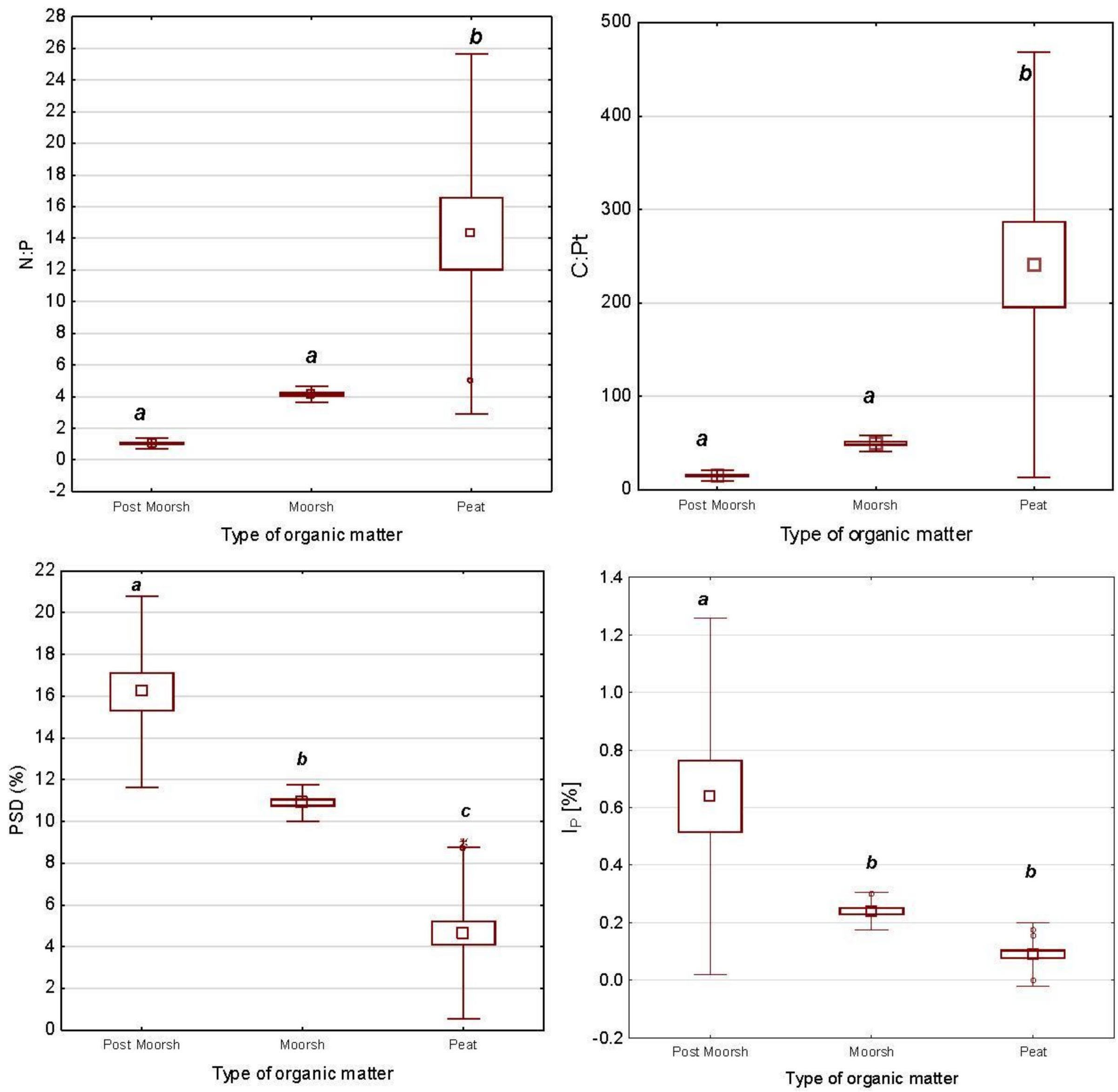


**Figure 6.** Variability range of calculated parameters (PSD, C:P, N:P, Ip)[1] in soil organic horizons categorised by type of organic matter for: Post Moorsh (A horizons of P1), Moorsh (M horizons of P2), Peat (O horizons of P2 and P3). Explanation: central squares—mean values; boxes—mean values ± standard errors; whiskers—mean values ± 2*standard deviations; circle—outliers; asterisk—extreme; letters a, b, c—significance of differences (homogeneous groups of means) by RIR Tukey test (at $p < 0.05$). [1] Pt—total P; PSD—P saturation degree; Ip—P solubility index.

A logical consequence of P retention in soils as a result of OM transformations is the positive correlation (statistically significant for $p > 0.05$) obtained between the PSD index and most of the examined P forms (Pt, Pmin, Pox, Porg) in the soils studied (Table 6). The significant strong positive relationships between the PSD and soluble P forms (Pw, $P_{CaCl2}$ and $P_{M3}$) denote the importance of this indicator in predicting easily mobile P forms not only in mineral soils but also in organic soils.

### 3.4.2. Ratios of C:P and N:P

The C:P ratio is also considered to be a good indicator of organic matter transformation in organic soils. The changes of C:P (as well as the C:Porg ratio) are very pronounced (from 356:1 to 12:1) and strongly reflect the release of P along with OM mineralisation and decreasing C content in the organic horizons of the studied soils (Table 5). The C:P ratio values decrease in the order P3 > P2 > P1 (Figure 6, Table 5). The N:P ratio is of

similar importance in assessing OM and P transformations, which decreases from a value of about 10–20:1 in peat soil (P3) to 4:1 in murshic soil (P2) and reaches a ratio of 1:1 in the postmurshic soil (P1). The significance of these changes was verified statistically (Table 6) and the variations in this ratio are presented in graph form (Figure 6).

Strong, statistically significant correlations were calculated between the C:N, N:Pt, C:Pt ratios and Pt, Pox, Pmin and soluble P forms (Pw, $P_{CaCl2}$, $P_{M3}$), as well as with Ash and C content. These findings confirm that OM changes play an important role in terms of the P distribution in the organic horizons of the studied soils (Table 6). Most of these correlations are also statistically significant for Porg; this index reflects the stage of Porg release from plant tissues, conversion of Porg from the decomposition of organic components into mineral forms, and OM mineralisation. Thus, all the relationships confirm significant linkage between OM transformations, indicators tracing these processes, and P changes in the studied soils. Our results indicate likely P release into soil solution (and potentially into the environment), especially within organic horizons which remain in oxidative conditions, where OM transformations occur intensively.

#### 3.4.3. Solubility Index (Ip)

The solubility index (Ip) is often used in assessments of OM transformations (e.g., [86]), however, it is also more broadly a reliable indicator of P transformations. The Ip value reflects the intensity of the soluble P forms released at the studied site. Its usefulness is confirmed by the results of our study, which clearly show a gradual increase of Ip value in the order P3 < P2 < P1. This relationship is also observed in vertical sections of the organic horizons within the same soils, namely, Ip values increase with the intensity of drainage degree and OM transformation (Figure 6). The strong statistically significant positive correlation coefficients between Ip and $Pw/P_{CaCl2}/P_{M3}$ confirm the validity of these relationships (Table 6). This demonstrates that the Ip parameter can be used to estimate P release in organic soils. Furthermore, its strong correlation with soil properties (Table 7) which reflect organic matter transformations (i.e., Ash, C, N, C:Pt, C:Porg, and N:Pt) confirms the potential usefulness of Ip in estimating the extent of organic matter transformation in organic soils. The Ip index shows some similarities with the PSD parameter (Table 7), which represents the degree of P saturation of soils and whose value strongly depends on the content of Alox, Feox and Pox, which is a key limitation of the PSD index. The strong positive correlation between PSD and Ip (r = 0.89 for $p < 0.05$) and the similarity between Ip and PSD in their relationship with other parameters (Tables 6 and 7) suggests the possibility of using Ip and PSD to assess the degree of OM transformation and solubility of P forms in organic soils under variable drainage conditions, indicating the state of the soil degradation.

**Table 7.** Correlation coefficients for Ip and PSD indicators and Ash, C, N and N:P, C:P ratios. Correlation coefficients significant at $p < 0.05$ are shown in bold.

| Variable | Ash | C | N | N:Pt | C:Pt | C:Porg | Ip | PSD |
|---|---|---|---|---|---|---|---|---|
| Ip [1] | **0.73** | **−0.75** | **−0.77** | **−0.65** | **−0.62** | **−0.67** | - | **0.89** |
| PSD | **0.94** | **−0.95** | **−0.94** | **−0.87** | **−0.84** | **−0.88** | **0.89** | - |

[1] Pt—total P; Porg—organic P; Ip—P solubility index; PSD—P saturation degree.

### 3.5. Principal Component Analysis (PCA)

The PCA diagram displays the relationship between soil organic horizons in studied soils (P1–P3) and suggests which parameters potentially best characterise each sample. In a two-dimensional representation, each axis explains a certain percentage of the total variability that exists between samples. Figure 7 shows the distribution of the parameters measured in the soil samples in this study in terms of their relationship to principal components 1 and 2 (PC1 and PC2). PC1 and PC2 explained 79.13% (59.63% + 19.50% from PC1 and PC2, respectively) of the total variability among the selected properties of organic horizons; this validates the two-dimensional PCA representation approach to describe

the sample characteristics. The C:P, C:Porg, N:P, and Alox parameters contribute most to PC1, while the pH, Feox and $CaCO_3$ parameters make the largest contribution to PC2 (Figure 7A). In Figure 7A, parameters with longer vectors better explain the variability in properties among organic horizons than the shorter vectors. Alox, $CaCO_3$ and N are represented by shorter vectors in the PC1–PC2 coordinate space, suggesting that these parameters are less strongly correlated with overall sample characteristics (Figure 7A). Vectors closer to each other indicate parameters that may have a strong positive correlation, whereas vectors that form an angle closer to 180° may have a negative linear correlation. A number of important groups are positively correlated in Figure 7A. These include Ash and PSD, which are negatively correlated to N and C. These parameters show the basic changes in soil properties related to the organic matter degradation process, i.e., increasing ash content, reduced content of organic carbon and nitrogen, and growing P saturation degree. Positive correlations are observed between Porg (as a fraction of Pt) and C:N, C:Pt (C:Porg) and N:Pt; these parameters are also negatively correlated to Pt, Pox, Pmin and Porg (weaker correlation). These values indicate general changes in soil material, indicating P is an important element in terms of OM changes. The lowering percentage share of Porg and decreasing ratio values of C:N are strongly related in terms of principal components, and C:P (C:Porg) and N:P are also significantly related to increasing content of P (Pt, Pox, Pmin, Porg) along with organic matter mineralisation under altered humidity conditions in soils. The positive correlation between soluble P forms and solubility ratio (i.e., Pw, $P_{CaCl2}$, $P_{M3}$ and Ip) is also indicated in Figure 7A; in addition, the soluble P forms are positively correlated with PSD and Ash in principal component space.

The position of the soil organic horizons displayed in Figure 7B indicates four visible separated groups. The first represents peat horizons in P3 that are submerged in water in reductive conditions, therefore, this cluster represents organic matter in the accumulative phase. The second group corresponds to the Oe horizon of the same soil, i.e., the peat surface horizon affected by aerobic conditions; this cluster is therefore dominated by the process of mineralisation of organic material. The third cluster correlates to moorsh horizons in murshic soil (P2), including the peat horizon (Oa) lying beneath the moorsh layers (M1–M3). The spatial clustering of all the organic horizons (M and O) of P2 into one group indicate that despite the differences in OM character, their overall properties allow them to be classified together into the same group. Some of the basic soil properties (e.g., $CaCO_3$, pH and Feox content) fall within the same quadrant (Figure 7A). The content values for different P forms (i.e., Pt, Porg, Pmin, and Pox) represent a second group of parameters that lie within this quadrant in PC1–PC2 space. This is in good accordance with other observations; specifically, the content of total and mineral P was observed to increase in P2 compared to P3, and the reduction in Porg values is explained by intense mineralisation and degradation processes in these horizons. The fourth group (Post Moorsh horizons) corresponds to the organic horizons (Au) in the postmurshic soil (P1). This cluster lies in the same quadrant as Ash, PSD and Pw, $P_{CaCl2}$, $P_{M3}$ and $I_p$; these features show the most pronounced differences in P1 relative to P2 and P3.

The PCA allows different groups of soil organic material to be discriminated based on their character and degree of transformation of organic matter in relation to humidity conditions in the soil. This analysis identified the Oa horizons in P3 as a distinct group, for example, but the Moorsh and Peat horizons of P2 were clustered into one group. This indicates the greater importance of specific basic soil properties (pH, $CaCO_3$, Feox) in this soil rather than the degree of OM transformation.

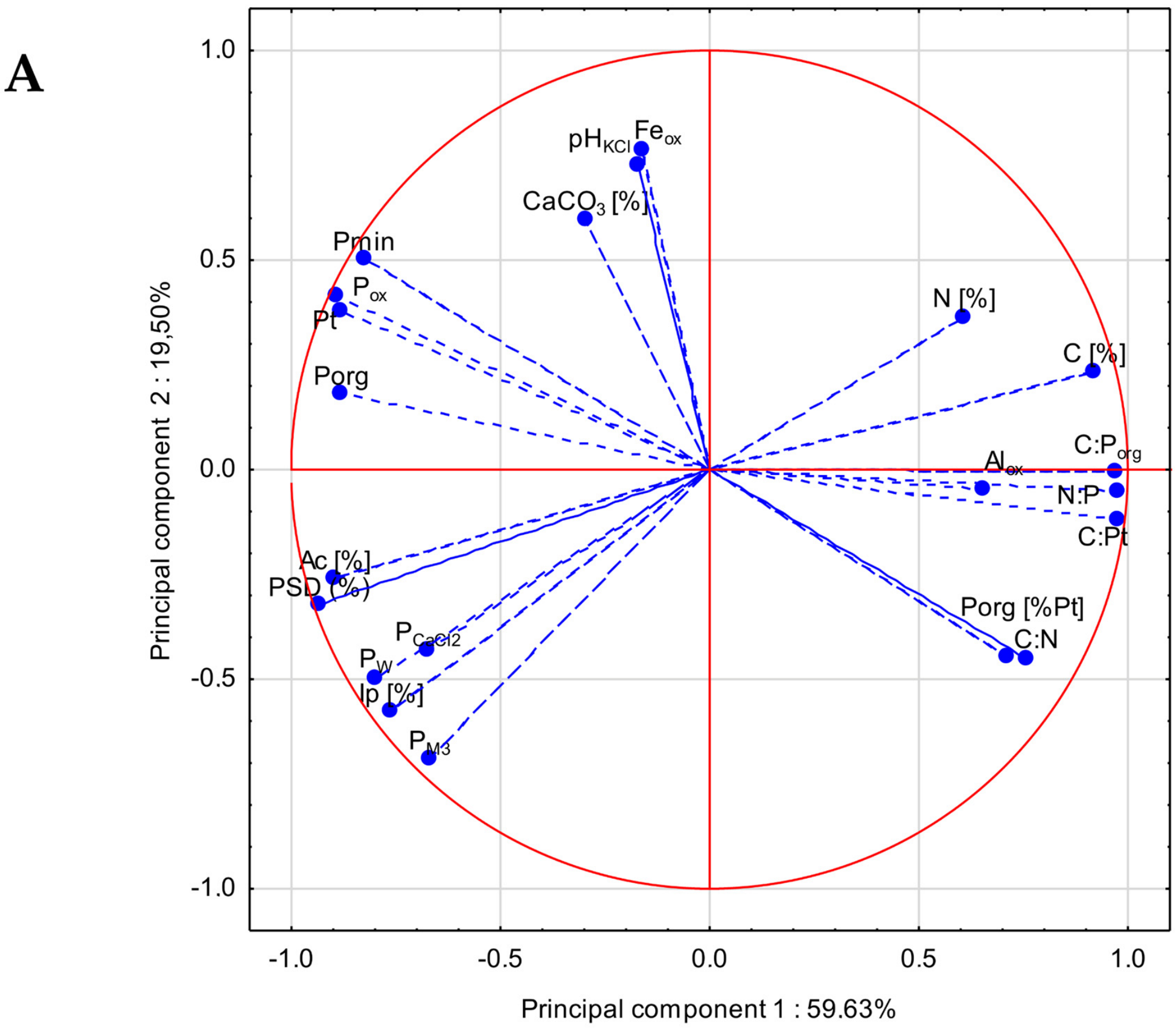


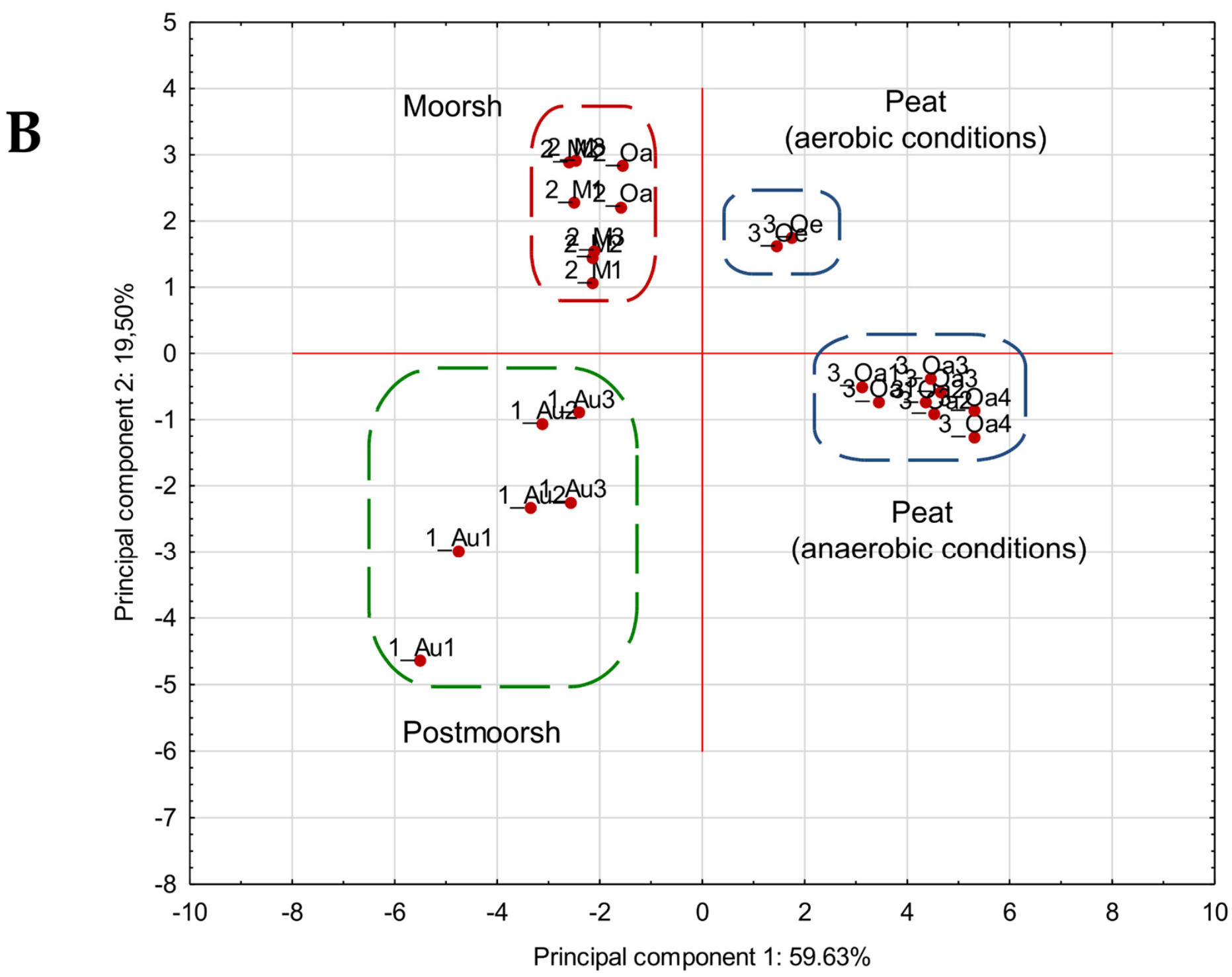


**Figure 7.** Principal component analysis (PCA) diagram of variables (**A**) and cases (**B**) for organic horizons of soils under study.

The transformation of drainage-sensitive organic soils causes many changes during the organic matter transformation. The results of the study confirmed that degradation of organic soils may pose a significant ecological threat related to potential release of P into the environment, which may have a negative impact on the quality of water reservoirs by accelerating their eutrophication. This phenomenon may be particularly dangerous when it occurs in the vicinity of cultivated soils. An important challenge in the context of the discussed problem is to minimise the degradation processes of drained organic soils, especially peatlands. Although peat formation is a slow process, rewetting a degraded area may be the most appropriate way to reduce $CO_2$ (and other greenhouse gases) emissions from peat oxidation [87]. Rewetting can enable drained peatlands to become peat-forming ecosystems and turn them again into sinks for soil carbon and nutrients (including P, which—in the course of long-term rewetting—is not only locked in organic matter but also transformed from labile P to stable P fractions at the surface horizons of the different peatland types) [88,89]. The goal of peatland rewetting is to achieve a persistent and stable saturation of the peat body with water by raising the water table to the peat surface and reducing the water level fluctuations [90].

## 4. Conclusions

The results of our study on fen soils located in the Odra river valley showed changes in the chemical properties of organic soils degrading as a result of reduced moisture content. The most important environmental problem at the studied sites is related to the accumulation of easily mobile P forms. This is confirmed by significantly increased content of soluble P forms (Pw, $P_{CaCl2}$, $P_{M3}$), particularly in the most drained postmurshic soil (P1). The indices used in this study—Ip, PSD, C:Pt, N:Pt—reflect well the P and OM transformations in organic soils degraded by drainage. This is indicated by numerous statistically significant relationships between the indices and basic soil properties (e.g., Ash, C, N), as well as different P forms (Pt, Pmin, Pox, Porg, Pw, $P_{CaCl2}$, $P_{M3}$). The PSD and Ip values increase and the C:Pt and N:Pt ratios decrease with the degree of OM mineralisation and the degree of site drainage (P3 < P2 < P1). The PCA analysis confirmed the above observations, indicating close relationships between the different groups of parameters and separating the organic horizons into four groups. This supports the advancement of the degradation process and the transformation of organic matter and P, as well as the variability of other soil properties. It is advisable to use these indicators in further studies on organic soils and macro element transformations under degradation conditions.

**Supplementary Materials:** The following are available online at https://www.mdpi.com/article/10.3390/agronomy11101997/s1, Table S1: Degree of organic matter decomposition in organic horizons in studied soils, Table S2: Sorptive properties of soils, Table S3: Percentage share of P forms [in % of Pt content], Figure S1: Colour-coded plot showing results of grouping objects and soil features (soluble P forms: Pw, $P_{CaCl2}$, $P_{M3}$).

**Author Contributions:** Conceptualisation, M.D. and A.B.; methodology, M.D. and A.B.; formal analysis, M.D.; investigation, M.D., A.B. and K.K.; resources, M.D., A.B. and K.K.; data curation, M.D.; writing—original draft preparation, M.D.; writing—review and editing, M.D. and A.B.; visualisation, M.D.; funding acquisition, M.D. All authors have read and agreed to the published version of the manuscript.

**Funding:** This work was financed by the statute project of Wrocław University of Environmental and Life Sciences, Institute of Soil Science and Environmental Protection from a subsidy of the Ministry of Education and Science of Poland.

**Institutional Review Board Statement:** Not applicable.

**Informed Consent Statement:** Not applicable.

**Data Availability Statement:** The data supporting reported results is available from the corresponding author upon request.

**Acknowledgments:** The authors highly appreciated the assistance and technical support received from Angela Potoniec, Szymon Jędrzejewski, Jarosław Szadorski, Institute of Soil Science and Environmental Protection, Wrocław University of Environmental and Life Sciences.

**Conflicts of Interest:** The authors declare no conflict of interest. The funders had no role in the design of the study; in the collection, analyses, or interpretation of data; in the writing of the manuscript, or in the decision to publish the results.

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
