# Peer review of "Phosphorus Behaviour and Its Basic Indices under Organic Matter Transformation in Variable Moisture Conditions: A Case Study of Fen Organic Soils in the Odra River Valley, Poland"

_agronomy, doi:10.3390/agronomy11101997_

Round 1
Reviewer 1 Report
I found the work interesting from a pedological point of view. However, I have some doubts about the agronomic value of the manuscript.
Indeed, no data were shown on differences in terms of flora or crops of agronomic value.
Furthermore, the authors discuss little about the hydrographic aspects of the territory.
In which season were the measurements taken? What fungal population was present in the soil?
In general the manuscript is descriptive and no hypotheses or experimental comparisons are reported.
For this reason I suggest a thorough revision of the manuscript.
Reviewer 2 Report
please modify the manuscript according to the suggestions

Reviewer 3 Report
Manuscript titled: Phosphorus behaviour and its basic indices under organic matter transformation in variable moisture conditions: a case study of fen organic soils in the Odra river valley, Poland prepared by Debicka et al. presents very interesting results and valuable data. I appreciate the topic and main aim of the research. However, I have found some discrepancies and details that needs to be completed.
- english language has to be improved
- Abstract needs to be re-styled- please add details of results
- Introduction- some informations are too general and not required - remove pharagraph 3 - lines 46-59.
- Some arguments need to be corrected - line 130- P2 and P3 areas are fed by rivers or groundwater - please specify
- Figure 1- improve quality
- Figure 1 - graph is not readable (x and y axis)
- Methods (part 2.1) -soil sampling description is fully required
- Soil samples were sieved with biological (plants) residues ? Please specify
- For analysis fresh soil samples were used. however I did not find information about soil homogenization and how was obtained the representative soil sample from each soil for chemical analysis?
- Add reference or describe Sheibler method for carbonate analysis
- Please add information how did you distinguish organic carbon from total carbon by Varion Macro macroanalyser. Additionally add producer and country to instrument
- change M to mol/l in whole manuscript
- Charcterize Mehlich III extraction agent (line 199) or use reference for extraction protocol
- Table 2 -Ash content - use another abbreviation as Ac (activated carbon)
- Table 2 - results of all pH determinations need to be clarified. Is the presented value equal to average value of replicates measurement? Or is it single measurement? The standard deviations are fully required.
- Please correct description of statistical analysis (p or P value ?)
- Figure 4 - improve quality
- conclusions - more details and obtained results are required
